# Women's agricultural practices and their effects on soil nutrient content in the Nyalenda urban gardens of Kisumu, Kenya

Nicolette Tamara Regina Johanna Maria Jonkman[1], Esmee Daniëlle Kooijman[1], Karsten Kalbitz[2], Nicky Rosa Maria Pouw[3], Boris Jansen[1]

[1] Ecosystem and Landscape Dynamics group, Institute for Biodiversity and Ecosystem Dynamics, University of Amsterdam, Amsterdam, 1090 GE, the Netherlands

[2] Soil Resources and Land Use, Institute of Soil Science and Site Ecology, Technische Universität of Dresden, Tharandt, 01737, Germany

[3] Governance and Inclusive Development Programme Group, Amsterdam Institute for Social Science Research, University of Amsterdam, Amsterdam, 1018 WS, The Netherlands

*Correspondence to*: Nicolette Tamara Jonkman (N.t.Jonkman@uva.nl)

**Acknowledgements**

We would like to sincerely thank all those who aided us in the preparation and execution of this research project. Among whom: the Kisumu VIRED team, including Professor JB Okeyo-Owour and Dr. Dan Abuto, the CABE team in Nairobi, including Dr. Hannington Odame, and NWO-WOTRO for funding of the project (W 08.250.200).

**Abstract.** In Kisumu up to 60% of the inhabitants practices some form of urban agriculture, with just under 50% of the workers being female. On average, women spend more hours a day in the gardens than men. Therefore women's knowledge is pivotal for effective agricultural management. To enhance and better use women's knowledge, gender related social cultural obstacles linked to land ownership, investment, and farm inputs have to be taken into account. We aimed to determine how the agricultural knowledge and motivations of women farmers working in the Nyalenda urban gardens in Kisumu (Kenya) influence the soil nutrient status as reflected by the total soil C and N, available soil N and P and exchangeable soil Na, K, Mg and Ca. Two prevailing practices were compared to determine how the agricultural management practice influences soil nutrient content: 1) applying manure only, and: 2) applying manure while intercropping with cowpeas. Interviews and focus group discussions were held to explore the agricultural knowledge and motivations of the women working in Nyalenda. Soil analysis showed that the soil in Nyalenda was rich in nutrients overall, but that the intercropped fields contained significantly lower total soil nutrients than fields where only manure was applied. While theoretically intercropping with a legume such as cowpeas should increase soil N content, due to socio-economic factors, such as poverty, intercropping was applied in a way that did not increase soil nutrient contents; rather it diversified revenue. The knowledge of the women farmers was found to be limited to practical and sensory knowledge. This shows that when aiming to improve soil nutrient status and agricultural yields through agricultural training, socio-economic conditions and cultural context, as well as gendered knowledge differentiation has to be acknowledged.

**Keywords**: Agricultural management, gender and knowledge, urban agriculture, urban horticulture, soil nutrient content

**Introduction**

This paper builds on a case study conducted in the urban gardens of Kisumu, Kenya. The study seeks to link women's knowledge on agricultural practices and their motivations in choosing specific practices with the nutrient content of their soils at this site. Urban gardening is part of the practice of urban agriculture, which encompasses all agricultural activities practiced within municipal borders (FAO, 2012). Urban gardening is found in most cities in the developing nations, such as Kenya, and ranges in scale from sack gardening to several acres being used for food production. For the urban poor the urban gardens provide employment and affordable vegetables. These vegetables are less expensive than those imported from the rural areas due to the lack of transporting costs. Moreover, limited infrastructure makes it difficult for fresh vegetables to be transported from the rural areas. By growing such vegetables within the municipality the costs are reduced and the lower costs of these vegetables allow the urban gardens to contribute to urban diet diversity and food and nutrition security (FAO, 2012; Gallaher et al., 2013). Urban gardening is a possible way to increase local food and nutrition security and provide employment. However, there are concerns surrounding urban gardening, including some concerning health risks and environmental degradation (Cofie et al., 2003; FAO, 2012).

With over half a million inhabitants Kisumu is Kenya's third largest city. Unemployment in Kisumu is high; in 2013 the unemployment rate in Kisumu was 40% (Mireri, 2013). Over 60% of Kisumu's inhabitants live in informal settlements (Mireri, 2013; Obade, 2014; UN-Habitat, 2005). An estimated 60% of the inhabitants of Kisumu practice some form of urban agriculture, including livestock keeping. Agriculture has been practiced on the periphery of the city since its founding in 1901, but as the city grew the boundaries between the urban areas and the rural areas have faded. The agricultural areas were fragmented and now fall within the municipal boundary. As such, these areas have been reclassified as urban gardens (Anyumba, 1995). The urban gardens are located on the edges of the informal settlements.

Mireri (2013) found that approximately 47% of those working in the urban gardens in Kisumu is female, and that on average women spend more hours a day on the farms than men. Women in Kenya and many other countries in sub-Saharan Africa are culturally expected to take responsibility for household duties like family food provisioning, cleaning the house, and caring for children and elderly people. As urban agriculture sites in most sub-Saharan cities are relatively near the home it is an accessible option to women who also have many other daily tasks (Doss et al., 2018; Mougeot, 2000; Poulsen et al., 2015; Simiyu and Foeken, 2013). Many of the women farmers in Kenya (Kabira, 2007; Kameri-Mbote, 2006; Kiriti-Ng'anga, 2015a; Kiriti-Ng'anga, 2015b) and in Kisumu in particular (Likoko et al., 2019; Mireri, 2013) work on a subsistence basis and any excess produce is sold to pay for expenses such as their children's education (Mireri, 2013; World Bank, 2009). Gender inequality makes it difficult for these women to move beyond subsistence agriculture. Women in Kenya and in most of sub-Saharan Africa face greater obstacles than men in regards to land ownership, investment, and farm inputs due to historical, social-cultural, and financial constraints as a consequence of their gender (Alunga and William, 2013; Dolan, 2015; Kabira, 2015; Kameri-Mbote, 2006; Kiriti-Ng'anga, 2015a; Kiriti-Ng'anga, 2015b; Likoko et al., 2019). As a consequence, few modern techniques are applied in the urban gardens of sub-Saharan Africa. Lack of access to capital, knowledge, and security limits these women to traditional techniques and sensory knowledge passed down within families and small social circles (FAO, 2012).

Overall, women consistently have agricultural yields that are on average 20-30% less than men in developing countries, due to a lack of equal access to technologies such as fertilizers (FAO, 2006). At the same time, there is evidence of gender differentiated access to knowledge. The results of a food security survey held by the African Women's Studies Centre and the Kenyan National Statistics Bureau in 2013 in several counties in Kenya showed that women respond differently to food security issues and consider challenges differently than men. For example, up to 80% of men believe that a small,

uneconomical area of land is a hindrance to achieving food security, whereas only 20% of women consider this to be a major hindrance (KNBS/AWSC, 2014*)*. Research by Saito et al., 1994 suggests that women could potentially produce up to 20% more on the same surface area than men if given equal access to resources in sub-Saharan Africa. However, there is also research showing that due to a lack of education and training, women farmers use practices that are less environmentally

friendly and can lead to a more rapid degradation of the soil (Doss et al., 2018). It could be expected that sources of information and knowledge are present and accessible more readily in an urban environment, and that because of this the women in urban agriculture could be more knowledgeable on agricultural management practices than their rural counterparts. However, when farmers do not use scientific findings it is often regarded as a sign of unwillingness, lack of understanding, or ignorance. This view is particularly damaging for the collaborative interactions between different

institutions and farmers, and the success of any potential innovations in agriculture that are adaptive, affordable, and applicable to the context.

Therefore, we aimed to determine how the agricultural knowledge and motivations of women farmers working in urban gardens in Kisumu (Kenya) influences their soil's nutrient status as reflected by the total soil C and N, available soil N and P

and exchangeable soil Na, K, Mg and Ca in their urban agricultural plots. The methods used combine analyses from the natural and social sciences and were designed to triangulate and provide complementary information. We specifically aimed to study women farmers as an important group in their own right rather than making a comparison with men as farmers.

**Materials and Methods**

The decision to work with the Mesopotamia group was made after various meetings with partners from a local scientific

institution and NGO[1] in conference with the Mesopotamia group itself in January 2016. The Mesopotamia group was seen by the scientists gathered at the conference as representative for many of the urban gardening groups in Kisumu, and especially those working on the border of the Nyalenda informal settlement. The site is considered to be representative of the area and farmers groups in the urban gardens of the city in terms of size and composition, as well as the prevalent soil type and water accessibility at the site itself. Nyalenda is one of six informal settlements in Kisumu and is one of the largest, both

in number of inhabitants and surface area covered (UN-Habitat, 2005). Divided over two blocks, A and B, Nyalenda houses nearly 50.000 people within an area of 8.1 $km^2$. Existing infrastructure, access to electricity, and access to sanitation are limited in the informal settlements (UN-Habitat, 2005). All along the southern edge of Nyalenda there are active vegetable farms adjacent a river and wetland area. One of the groups active in these urban gardens is the Mesopotamia group. The group consists of 14 members, 8 women and 6 men, who cultivate an area of 3-4 ha. Most Mesopotamia members have

inherited their land and some rent extra plots within the area; the group is diverse in age and experience. The farming group is characteristic for the urban gardening situation that can be found throughout other cities of Kenya and (sub-) tropical Africa in general.

The Mesopotamia group had previously been informed by government extension services that their soil might be lacking in nitrogen (N). In response to this apparent lack of N at least 5 group members changed their practices, they started to

intercrop the local staple crop Sukuma Wiki, a kale (*Brassica oleracea var. Sabellica*) with a legume with nitrogen fixating root nodules, cowpeas (*Vigna unguiculata L. Walp*) in 2013 (Likoko and Jonkman, 2016).

The research approach combines semi-structured interviews and focus group discussions with women food entrepreneurs (WFE's) working in Nyalenda, and soil analysis of their urban garden plots. The four fields selected for soil sampling were

---

[1] Victoria Institute for Research on Environment and Development (VIRED), Centre for African Bio-Entrepreneurship (CABE),

all used to grow kales, in two of the fields the kales were intercropped with cowpeas. All four sampled fields are centrally located in the urban gardens, limiting the differential influence the nearby river might have on fields lying closer or farther away from it. The soils of these fields were classified as Vertisols (FAO, 2014), characterized by the presence of heavy clay which shows shrinking and swelling behaviour. All samples were collected in May during the dry season. On each of the four fields 12 samples were collected from the topsoil (0-15 cm) to limit the influence of spatial variability, 48 samples total. All samples were subsequently dried at $70^0$C, sieved at 2 mm and stored for analysis.

**2.1 Interviews and Focus Group Discussions**

The four fields sampled are owned by two female members of the Mesopotamia group, each member owning two of the fields. The two women that own the sampled fields and 6 other female members of Mesopotamia were interviewed to determine their agricultural knowledge and management choices. The eight women varied in age and experience (table 1), capturing a broad spectrum of views and knowledge. The semi-structured interviews used open questions to determine what knowledge women farmers had about the effects of fertilizers on crops and soil, where they obtained this information, and to what degree and with whom they shared this knowledge. A structured questionnaire was used for the interviews to gather complementary and comparable information on the women's knowledge and views. There were two themes incorporated in the interviews, fertilizer use and information gathering and use. The questionnaire can be found in the supplement of this article, section S1. The interviews were conducted with the assistance of an interview guide, including an introduction, opening questions, key questions and a summary (adapted from Woodhouse, 1998; Curry, 2015a).

Table 1. list of interview participants including approximate age, number of fields (size between 0.1 and 1.5 acre), and farming experience in years.

| Interviewees: | | | | |
| --- | --- | --- | --- | --- |
| ID | Gender | Age (approximate) | # of fields | Farming experience |
| I1 | female | early 40's | 3 | 5-6 years |
| I2 | female | late 40's | 3 | 8 years |
| I3 | female | late 20's | 3 | 5 years |
| I4 | female | early 30's | 6 | 5 years |
| I5 | female | late 30's | 3 | 8 years |
| I6 | female | late 20's | 4 | 6 years |
| I7 | female | late 30's | 4 | 13 years |
| I8 | female | early 50's | 8 | 15 years |

In addition to the interviews, two focus group discussions (FGD) were held with members of the Mesopotamia group. One focus group discussion was held with 6 female participants and another with 11 participants, 6 women and 5 men. A women's only discussion was held with the 6 women participating to go more in depth on the knowledge of women, given the aim of our study to focus on women farmers as a group in its own right. The focus groups discussions were based on questions used in the interviews and the methodology proposed by Curry (2015b), Krueger & Casey (2002) and Johnson & Mayoux (1998), which may be found in the supplements of this article (S2). The discussions were aimed at determining the extent of agricultural knowledge in the Mesopotamia group as well as their information sources and the relative importance of these to the farmers. Both focus group discussions had the same format and started with a short opening and introduction followed by an explanation of the goal and guidelines for the discussion. The opening was followed by a set of discussion questions and an exercise. The discussion was closed with a short summary by the discussion leader. Due to the open

platform and the presence of multiple participants the focus groups discussions provided more in-depth answers and clarifications, to support and complement the information from the interviews.

## 2.2 Laboratory analyses and data processing

The soil samples were analysed is to determine how soil nutrient contents was influenced by the management choices of the
women farmers. Water extracts of the soil samples were created (ratio 1:2.5) and used to determine pH and electrical conductivity (EC). These water extracts were then filtered and available P, K, S, Ca and Mg measured using a Perkin Elmer Optima 8000 ICP-OES Spectrometer. Available $NH_4^+$, $NO_3^-$, $PO_4^{3-}$ and $SO_4^{2-}$ in the extracts were determined on a Skalar SA-40 continuous-flow analyzer. Total organic and inorganic C in the extracts were measured using a Shimadzu TOC/TN analyzer.

 Filtered $BaCl_2$ extracts were used for the determination of exchangeable Fe, Mn, Mg, Ca, Al, and K with ICP-OES (Schwertfeger and Hendershot, 2009). Extracts were prepared using 100 ml $BaCl_2$ 0,125 M and 4 grams of sieved and milled soil sample (<2 mm). CEC was calculated as the sum of the values for exchangeable Ca, Mg, K and Na in $cmol_c/kg$.
Total C and N were determined with 50 mg of soil (<2mm, milled) by using a Elementar Vario EL cube CNS analyzer. Total
P, K, Ca, S and Mg were determined by measuring $HNO_3/HCl$ extracts with ICP-OES; extracts were prepared with 250 mg soil (<2 mm, milled), 6 ml HCl 37% and 2 ml $HNO_3$, and underwent microwave destruction (60 min;Tmax 220°C; Pmax 75bar). Total elemental composition of the soil samples was also determined using XRF analysis, using the Thermo Scientific XRF Analyzer Niton; setting: mining Cu/Zn; Standard: NIST 2709a PP 180-649; 160 seconds.

Variance within each field and between fields with different management practices was determined using an analysis of variance test. ANOVA was used in case of normal data distribution and Kruskal Wallis with non-normal data distribution (Burt et al., 2009). The strength and direction of the relationship between different parameters was determined using a correlation coefficient, Pearson's R. All statistical analysis was done in Matlab, version R2014b. The measured results and calculated variances where corroborated with the results of the interviews and focus groups discussions.

**Results**

### 3.1 Interviews

While the interviews started with enquiring into the typical daily activities these turned out to vary too much from person to person and season to season to provide a meaningful clustering. The interviews did show that a range of agricultural management practices is known and practiced within the Mesopotamia group. Specifically, the agricultural management
practices that the women spoke of during the interviews were: crop rotation, fallow periods, fertilization with manure, compost and mineral fertilizer, intercropping, and mulching. Of the interviewees 2 knew no other methods aside fertilization for improving the soil, 3 named mulching, 4 mentioned fallow periods, and 4 mentioned crop rotation. Three of the women found that they were limited in choice of management practice due to their socio-economic circumstances. For example, for these women fallow periods are not an option as their lands are simply too small. A certain yield is needed for sufficient
income generation and leaving the land or a portion of it fallow would mean a significant reduction in income. A lack of fallow periods combined with intense agriculture puts pressure on the land, which can lead to increased erosion and may result in diminished soil nutrient content and yield (*KNBS/AWSC, 2014*).

Fertilizer use is mostly in the form of locally produced or homemade compost (4 of 8) or the use of unprocessed cow manure
(2 of 8). The other 2 interviewees used either mineral fertilizer or cow manure with occasional application of mineral

fertilizer. All interviewees named fertilizer as something the soil needs for growing crops, but none really knew what fertilizer does for the soil in technical terms. The women indicated that their knowledge regarding the effects of fertilizer is limited to visible effects only. One told that she fertilizes when the plants will wilt slightly and the leaves start to yellow specifically. 6 of the interviewees told that they add extra fertilizer when their yields go down or when the plants grow less vigorously than usually, but 2 of these also mentioned other possible ways to increase production, like crop rotation or fallow periods. The agricultural information sources named during the interviews were relatives, including parents, grandparent, or husband, trainings by NGO's or extension workers, elementary school, and observing others. Information from relatives was most common, being named by 5 of the interviewees, followed by trainings by NGO's or extension workers, named by 3 interviewees. As point of interest, one of the interviewees got her information from her grandmother, who was also interviewed, and who got her information from training by NGO's and extension workers. The agricultural knowledge women accumulate over their lifetime thus travels in small family and social circles.

To support the soil analysis the two farmers, I2 and I7 (table 1), whose fields were sampled were interviewed more extensively than the other interviewees. Both farmers have at least 8 years of experience and principally grow kales. Both farmers use manure from cows and chicken mixed with organic waste. The farmer using only manure, I7, applies manure by ploughing it into the soil at time of planting and then applies manure again every 4-12 weeks as she feels is necessary. The farmer practicing intercropping, I2, applies manure as a topdressing at planting and approximately 8 weeks after planting. The cowpeas are broadcast on the field and a number of the plants are removed after 2 weeks to make room for the kales to grow. Both farmers plough using a handheld tool, reaching a depth of approximately 20 cm. Farmer I7 weeds and ploughs her field every 10 days, whereas farmer I2 weeds every 14 days and ploughs her fields only every 26 weeks. In the fields of farmer I7 kales were grown in both fields in the previous growing season. The fields of the intercropping farmer were left fallow for 6 months before planting the current crops; in one of the fields maize was grown before the fallow period. Both farmers I2 and I7 use occasional fallow periods lasting between 3 to 6 months. Farmer I2 using intercropping started this practice in 2013 and has been using it ever since. Farmer I7, using manure application only, has been applying this practice for at least five years.

Both farmers have had training by different NGO's and extension officers and incorporated this in their agricultural management, which has led them to decided to use different practices. Farmer I2 indicates however that knowledge is often quickly forgotten due to a combined lack of use by the farmers and a lack of follow-up visits from the NGO's/extension officers. This is reflected by her own adaptation of the intercropping technique. She indicates that she relies on observation of her crops and experience to determine if fertilizer is required. As her crops continue to grow well she does not feel the need to change her practice by ploughing the cowpeas into the soil, what they call green manuring, or changing her fertilization practices. The farmer only using manure, I7, learned about the use of manure and creating compost from an NGO. She indicated that using manure is better than leaving the land fallow as her crops continue to do well. The details of the sampled fields in terms of crops and management practices are provided in Table 2.

Table 2. Details on the 2 farmers whose fields were sampled with a description of their crops and agricultural management practice.

| | | Field type 1: Manure | | Field type 2: Intercropping + Manure | |
|---|---|---|---|---|---|
| **Farmer** | | I7, late 30's, 13 years' farming experience. Management practice established since 2011. | | I2, late 40's, 8 years' farming experience. Practiced intercropping since 2013. | |
| | | Field 1 | Field 2 | Field 1 | Field 2 |
| **Crops** | | Kales: Coverage 80% Average height 45cm | Kales: Coverage 85% Average height 90cm | Kales, intercropped with cowpeas: Coverage 60% Average height 50 cm | Kales, intercropped with cowpeas: Coverage 65% Average height 60 cm |
| **Agricultural Management Practice** | | | | | |
| **Planting** | | Kales (6 weeks) | Kales (28 weeks), Some of them were removed and new kales (1 week) were planted in between. | - Kales (11 weeks) - Cowpeas (4 weeks) | |
| **Fertilization** | | Compost from manure (cow, goat, sheep) and organic waste; applied with planting (6 weeks) | Compost from manure (cow, goat, sheep) and organic waste; applied with planting (28 weeks) and twice after (16 and 4 weeks) | - Compost from manure (cow, chicken) and organic waste; applied as topdressing after planting (9 weeks). - Intercropping: cowpea seeds were spread randomly (broadcasting) and most cowpea plants were removed from the fields 2 weeks after planting | |
| **Ploughing** | | Every 10 days | | Every 26 weeks | |
| **Weeding** | | Every 10 days | | Every 14 days | |

### 3.2 Focus group discussions

During the women-only FGD we learned that most of the women work in the urban gardens as a way of generating income, to provide for themselves and their children. At least half of the women participating in the FGD are widows and agriculture is their sole form of income. They do feel that their gender puts them at a disadvantage as they feel that there is a lack of mutual understanding between the men and women of the group and that the men have a tendency to refuse to share resources with them. They believe it would help if there would be at least 1 woman on the groups' board and that this would
lead to more equal distribution of resources among the group members, such as fertilizers received from a local NGO. There are some limitations on the women's activity due to cultural restriction, but not all of them are still actively followed. One limitation explicitly named during the women-only FGD still followed is the prohibition for women to plant and own trees. Banana trees for example can bring higher profits than some other crops.

Much of the information from the interviews was confirmed in the focus group discussions. The exceptions were as follows. Although the types of fertilizers named during the FGD were mostly the same as those in the interviews, during the mixed FGD more of those responding appear to be using a form of mineral fertilizer. This discrepancy may be due to the inclusion of men in this FGD. It also became apparent during the mixed FGD that the farmers do have knowledge of the way to work intercropping in a manner that can add N to the soil, in contrast to what came forward from the interviews, but that they have
a different name for this method: green manuring. During both FGDs and the interviews it came forward that the management of the farms is largely reactionary. Decisions regarding using one of the various methods to restore soil fertility are undertaken only when the crops seem to do less well than previous crops.

While mineral fertilizers seems to be used more than inferred from the interviews, the participants of the FGDs do show a preference for organic types of fertilizer. According to the Mesopotamia farmers vegetables grown with organic fertilizers
taste better and keep longer, whereas mineral fertilizer is reported to damage the soil.

When asked about the sources of agricultural information the participants of the mixed FGD named 5 different sources. All ranked family as the first and most important source, followed by trainings and demonstrations. Observation and visiting others was ranked third. Media and exhibitions were ranked fourth and fifth respectively, and the farmers indicated that this is because of their lack of access to media and the expenses involved in visiting exhibitions. This may indicate that accessibility remains a problem for these farmers, even though there are more and closer sources of information on agricultural management in the urban environment.

### 3.3 Soil Analysis

Intercropping is done by 4 of the women in the Mesopotamia group, mainly with cowpeas. Theoretically intercropping should improve soil nutrient content, specifically soil N content. However, the intercropping technique is not always applied in a way that would accomplish this: the cowpeas are harvested and not ploughed into the soil. Ploughing the cowpeas into the soil is needed for the nutrients accumulated to become available for other crops (Okalebo, 2009). The women also use intercropping to prevent soil erosion while the main crops, often kale, is still growing. Intercropping also provides a source of income while the farmer waits for the kale to mature as cowpeas mature faster.

The sampled soil was analysed for its nutrient content, and overall fell within the ranking 'high' by FAO standards (FAO, 2006). Table 3 provides the average values of the main soil parameters for all 4 sampled fields, as well as those parameters separated per management practice. The pH of the soil in the sampled fields ranged from neutral to very slightly alkaline, with an overall average of 7.3 (Table 3). The CEC was high overall with an average value of 34.0 $cmol_c$ $kg^{-1}$, likely as a consequence of the high clay content of the soil (Table 3). Similarly, with an average of 36.6 g $kg^{-1}$ the total soil carbon was also high. The laboratory analyses showed relatively high amounts of water soluble and exchangeable cations, however there is a significant difference in nutrient content depending on the management practice.

**Table 3. Average pH, total C and N (g/kg), exchangeable Mg, Ca, K and Na (mg/kg), CEC ($cmol_c$/kg) and water soluble ions $NO_3^-$, $NH_4^+$, $PO_4^-$ and $SO_4^{2-}$ (mg/kg) in the soil of Nyalanda field site overall, for fields with manure application only, and for fields with intercropping and manure application, standard deviations in parenthesis (0-15 cm depth, 4 fields, with 12 samples per field; n=48).**

| | pH | C | N | Mg | Ca | K | CEC | $NO_3^-$ | $NH_4^+$ | $PO_4^-$ | $SO_4^{2-}$ |
|---|---|---|---|---|---|---|---|---|---|---|---|
| | - | g $kg^{-1}$ | | mg $kg^{-1}$ | | | $cmol_c$ $kg^{-1}$ | mg $kg^{-1}$ | | | |
| **Overall average** | 7.3 (0.2) | 36.6 (11.0) | 2.8 (0.4) | 572.6 (89.2) | 4842.1 (761.2) | 1768.4 (879.2) | 34.0 (5.4) | 85.5 (62.7) | 5.9 (3.7) | 24.0 (15.6) | 59.0 (44.5) |
| **Manure only** | 7.3 (0.16) | 45.3 (8.9) | 3.13 (0.3) | 603.6 (102) | 5467.8 (467.9) | 2210.5 (1056) | 38.6 (3.2) | 96.7 (56.9) | 6.7 (4.9) | 31.2 (18.3) | 84.1 (56.9) |
| **Intercropping + Manure** | 7.1 (0.2) | 27.9 (3.4) | 2.6 (0.3) | 541.7 (60) | 4216.3 (395.9) | 1326.3 (200) | 29.5 (2.4) | 74.3 (66.3) | 5.0 (1.4) | 16.9 (6.9) | 33.9 (10.3) |

In case of manure application combined with intercropping the pH was neutral, whereas in case of only manure application the pH leaned towards very slightly alkaline (FAO, 2006). The CEC was nearly 10 $cmol_c$ $kg^{-1}$ higher in fields under only manuring than in the fields where there is also intercropping. Similarly, total soil carbon is nearly 20 g $kg^{-1}$ higher in the fields where only manure was applied in comparison with the fields where there was also intercropping (Table 1; Fig. 2a). There was no significant difference in soil organic carbon content between the intercropped and the manured fields (Fig. 2c). The contents of the macronutrients N, P, K, Ca and Mg were almost all higher under the field management type manure application only, as compared to manuring combined with intercropping (Fig. 1; Fig. 2).

Figure 1a, 1b, and 1c show the amounts of water soluble and exchangeable Mg, Ca and K as part of the total amount of the cation present in the soil, clearly demonstrating that the levels are higher under the practice of applying manure only. Figure 1d, 1e, and 1f show the proportion of the total amount of Mg, Ca and K in the soil that is water soluble or exchangeable.

5    Notable is that under the practice of manure application combined with intercropping, the average exchangeable fraction for Mg and Ca and the average water soluble fraction for Ca was higher, even though the absolute amounts are higher under manuring only (Fig. 1). The Kruskal Wallis and ANOVAs tests showed that all the described difference between the fields and between the management practices were significant for these characteristics at a confidence interval of 95%.

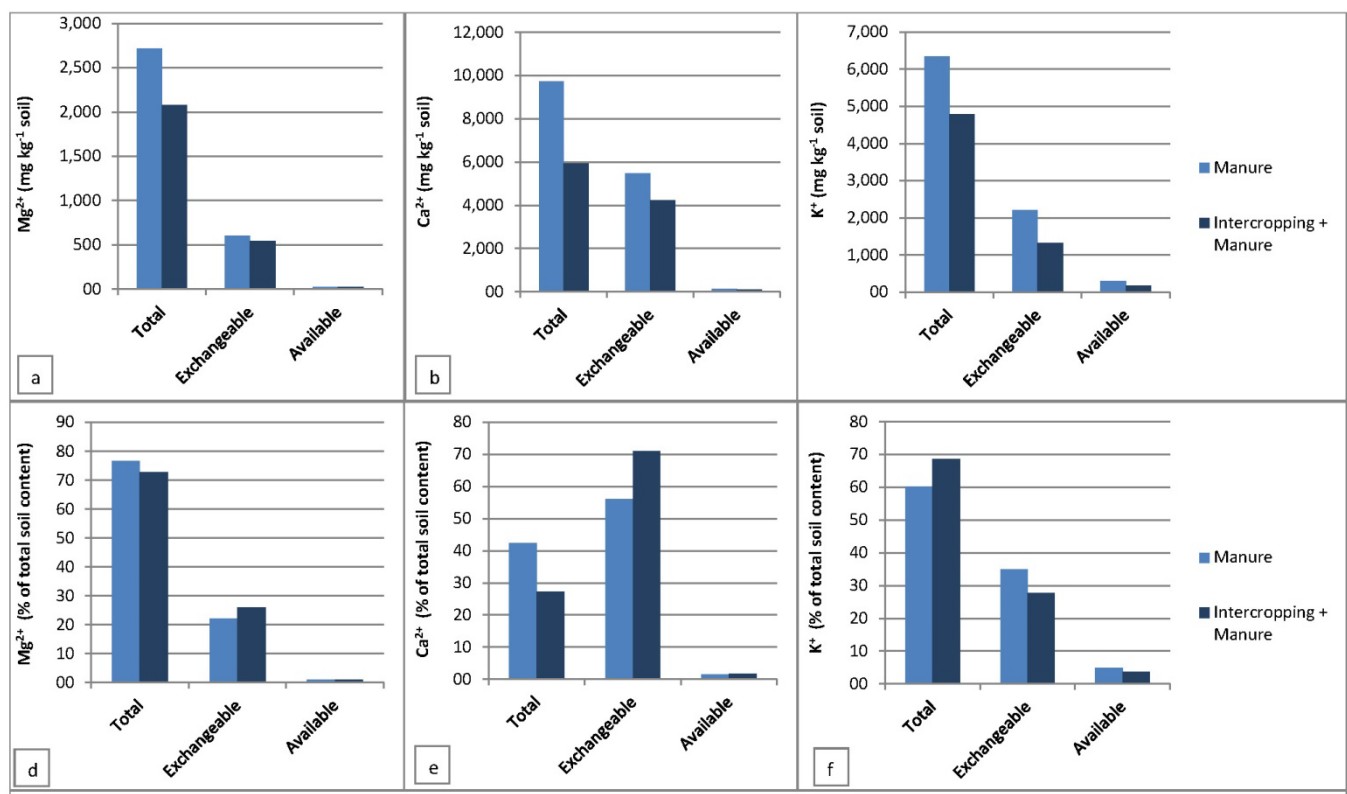

**Figure 1. 1a, 1b and 1c: Bars length show the total, exchangeable, and plant available/water soluble amount of Mg, Ca and K in mg kg⁻¹ soil under management 'manure' and 'intercropping and manure'. 1d, 1e and 1f: Total, exchangeable and plant available/water soluble Mg, Ca and K in the soil as percentage of the sum.**

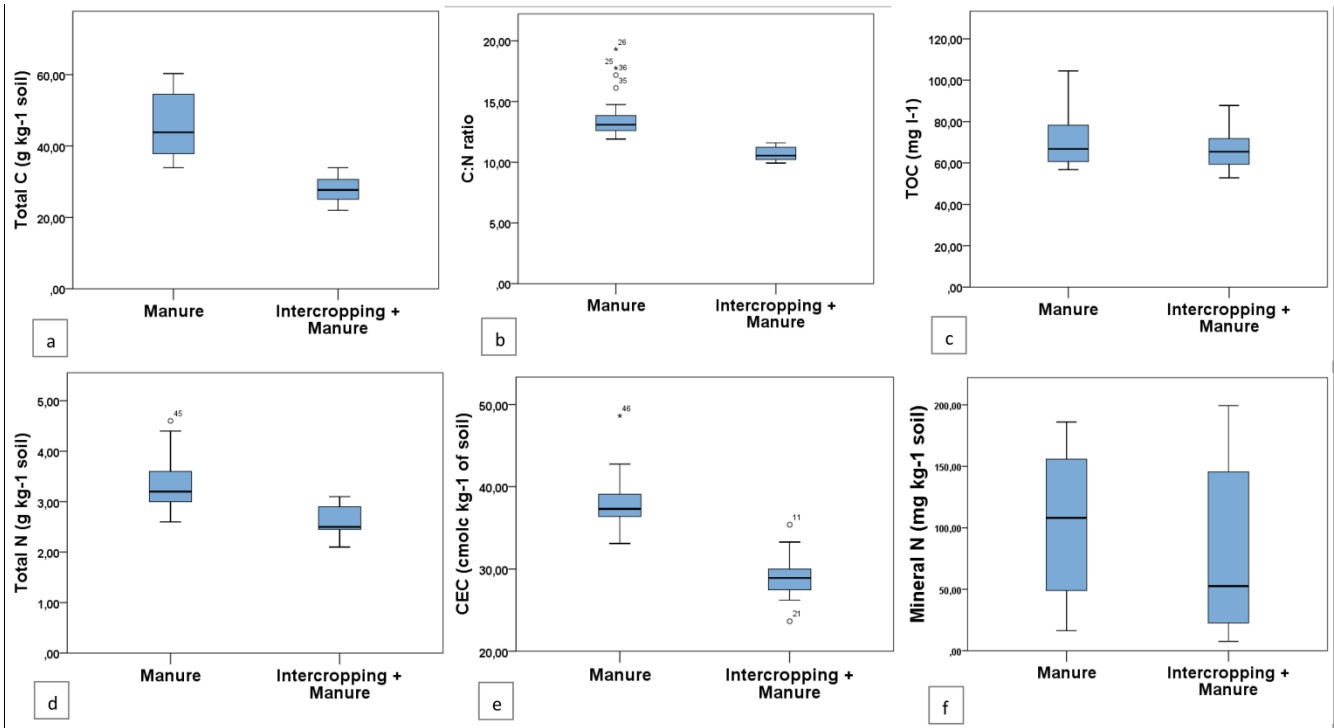

**Figure 2. Boxplots showing the differences between 2a: total C (g kg⁻¹ soil), 2b: C/N ratio, 2c: total organic C in the extracts (mg C l⁻¹), 2d: total N (g kg⁻¹ soil), 2e: cation exchange capacity (CEC in cmol$_c$ kg⁻¹) 2f: Mineral N (mg kg⁻¹ soil), for the agricultural management practices of 'manure' and 'intercropping and manure'.**

**Discussion**

Results of the sample analysis of the soil of the urban gardens in Nyalenda show a pattern consistent with the soil typology. The soils in the Nyalenda urban gardens can be classified as Vertisols (FAO, 2006; FAO, 2014). Vertisols are generally fertile and productive soils, high in Ca, K and Mg, but often poor in N and P (FAO, 2006, FAO, 2014). Soil analysis has shown the rating for exchangeable Ca and K was very high and the rating for Mg was high according to FAO classification (Table 3; Fig. 1) (FAO, 2014). The high amounts of nutrients in the Nyalenda soils is most likely because of the parent material, which consists of river and lake sediments with inputs from the African rift valley, and the limited age of the soil material.

Earlier research on soil nutrients and possible solutions for soil fertility problems in rural western Kenya concluded that socio-economic factors determine how likely it is that scientific findings and developments are taken up by farmers (Gicheru, 2012; Okalebo, 2005). The results from the interviews and the FGDs show that this is also the case in a more urban setting. This might be because the gardens were originally located in a more rural surrounding, but the urban centre of Kisumu has since spread to that area; effectively changing the gardens environs from rural to urban. However, the extension of the urban environment has not led to a greater accessibility to information for the farmers that now live in the informal settlements of the city. Information from media and exhibition were indicated to be too costly for the farmers in this study.

When asked to name and rank their primary sources of information all farmers of the Mesopotamia group indicated a preference for inherited knowledge, followed by trainings and demonstrations. Information from television or internet has less impact because these farmers lack access to these media. Exhibitions are considered good, but the expense to go and visit them is often considered to be too high. So even though the number of information sources is higher in the urban

environment, the access to these sources remains a challenge for the women farmers in Nyalenda that were part of the present study.

Limited access to sources of information means that most of the Mesopotamia farmers possess limited technical knowledge regarding soil processes, however they are aware of soil processes and their consequences in practical terms from sensory knowledge and daily experience. For example, the women are aware of the need to rest the soil with fallow periods or to do crop rotation, and that mulching improves soil structure. The majority of the women in the Mesopotamia group possess knowledge regarding agricultural management practices and the effects of these practices on soil in these basic terms. Knowledge is mainly passed down from previous generations or disseminates through observation of other farmers in the group or community. As such, agricultural knowledge generally stays within the confines of family and community groups within the urban gardens in SSA and does not travel far from its point of origin (Alunga and William, 2004; Kabira, 2007).

In Nyalenda, a context where poverty is widespread, agricultural management and decisions are heavily influenced by social-economic constraints. These constraints can work against sustainable farming practices. For example, the farmers explained that they took up intercropping with cowpeas originally because they were told that they needed to increase N in the soil by government extension workers. Analysis of the soil samples showed that the soil nutrient content is significantly lower when the kales are intercropped with cowpeas, while Mg and Ca showed a higher availability or are more readily exchangeable (Table 3, Fig. 1, 2). The cause for this difference is not clear, but may be the influence of the presence of a legume species. The presence of the rhizobacteria on the root nodules of the legume can promote the availability or exchangeability of nutrients beyond nitrogen by immobilizing nutrients and preventing them from leaching from the soil (Lavakush et al., 2014; Vejan et al., 2016). Furthermore, the lower amount of nutrients in the intercropped fields may be due to the different approaches that the farmers have to applying manure. The farmer who applies intercropping, I2, applies the manure as a topdressing only. The farmer who does not practice intercropping, I7, ploughs the manure into the field. Ploughing the manure into the field preserves N and promotes the biological breakdown of the manure, which increases the availability of the nutrients therein. Ploughing also results in greater contact between the soil mineral matrix and the organic matter, this promotes the stabilization of organic matter through binding to soil minerals and/or occlusion in soil aggregates (Baligar, 2001). The increased stabilization of organic matter in the soil could in turn have caused the significantly higher CEC in the fields of farmer I7. The decrease in soil nutrients in intercropped fields (Fig. 1) was most likely caused by not ploughing the cowpeas into the soil in combination with the different manuring practice (Okalebo, 2009). However, factors such as a difference in initial soil composition between the intercropped and non-intercropped fields prior to the adoption of intercropping may also have influenced the observed difference. Therefore, the observed differences must be interpreted with caution. Nevertheless, we feel we can conclude that three years of intercropping did not increase nutrient contents as was the original purpose of adopting the practice.

The interviews shed light on why the practice of intercropping is still continued if it does not lead to increased nutrient contents. Specifically, the interviews revealed that there is a sufficiently large financial incentive for the farmers to not change their current practice and that the farmers rely heavily on visible ques when deciding to apply a different practice or more fertilizer. So as long as the practice does not visibly influence the crops, farmers will not feel an urge to shift to another form of agricultural management. As the soil in the Nyalenda urban gardens is particularly rich (table 3), it is not likely that there will be a need for a large scale adaptation.

During the interviews it furthermore became clear that the farmers are harvesting the cowpeas for sale, instead of ploughing the cowpeas into the fields. Harvesting the cowpeas means a greater uptake of nutrients from the soil and no additional

organic material is added to the soil. Selling the cowpeas has become the primary motivation for intercropping as it gives the farmers a source of income in the period before the kales are mature and the advantage of doing so is more readily apparent to farmers than a potential increase of N in the soil. This shows that the lack of effect of intercropping on soil N contents in the examined soils is most likely not caused by a of lack of women vegetable farmers' knowledge of proper application or

the technical knowledge of intercropping. Rather it appears to be a conscious choice related to a shift in the aim to be achieved by intercropping, i.e. gaining a secondary crop to be harvested and sold rather than increasing the yields or quality of the primary crop.

## Conclusion

The results of the soil analysis showed that the soil in the Nyalanda urban gardens is rich in macronutrients. Further analysis indicated that, while seemingly small, there is a statistically significant difference between soil nutrients contents and availability between similar plots after 5 years of manuring only and 3 years of manuring plus intercropping with cowpeas. While nutrient contents is slightly lower in the intercropped plots, exchangeable nutrients Ca and Mg are slightly higher. The growing of the cowpeas beside the kales might be causing a more rapid extraction of nutrients from the soil. However, other

factors may also play a role, such as a different initial soil composition. Another difference between the practices of the two farmers whose fields were sampled was the fertilization scheme. The farmer that intercropped applied her manure as a topdressing, while the other farmer ploughed the manure into the soil. This too could have led to a difference in the total amount and availability of nutrients in the soil. Further research would need to be done to determine which of the two practices has the greater impact on soil nutrients: intercropping or fertilization method.

Nevertheless, even allowing for such other factors, the absence of any sign of increased nutrient content in the intercropped fields is remarkable, given that growing cowpeas and ploughing them into the soil should drastically increase soil N content. An explanation for this absence of an observed effect is that the intercropping farmer sold the cowpeas on the market rather than ploughing them into the soil. It is likely that the farmer originally took up intercropping to increase soil N after the advice from the visiting extension workers. However, she at some point changed her practice to make use of the opportunity

that harvesting the cowpeas offered her in terms of additional income. It is unclear when or why exactly she changed her practice, but it is likely that this was financially motivated. Due to the originally rich soil in the Nyalenda urban gardens, the farmer also saw no noticeable reduction in yield while intercropping.

The interviews and FGDs with the Mesopotamia group showed that there is knowledge present on a wide range of

agricultural management practices. However, the interviews with the individual women members of the group showed that the knowledge on these practices is unequally distributed and that while they may be known to a technique they do not possess technical knowledge on the effects of their management practices. We conclude that the incomplete knowledge of these farmers is a consequence of the way they acquire and rank knowledge, as well as their lack of access to alternative sources such as the internet. Further research would be needed to confirm this conclusion.


While this paper covers a case study of limited scale, meaning that this should be taken into consideration when viewing the results and drawing conclusions, the circumstances found within the Mesopotamia group are representative for many other groups in the urban gardens of cities in Kenya and subtropical Africa. The case study showed that women's decisions in agricultural management are influenced by their socio-economic and cultural status. Trainings should be adapted to take the

socio-economic circumstances of the trainees into account. Furthermore, the gender differences in ability and access should similarly be taken into account in order to improve the effectiveness of a given training or agricultural recommendations.

**Conclusion**

The results of the soil analysis showed that the soil in the Nyalanda urban gardens is rich in macronutrients. Further analysis indicated that, while seemingly small, there is a statistically significant difference between soil nutrients contents and availability between similar plots after 5 years of manuring only and 3 years of manuring plus intercropping with cowpeas. While nutrient contents are slightly lower in the intercropped plots, exchangeable nutrients Ca and Mg are slightly higher. The growing of the cowpeas beside the kales might be causing a more rapid extraction of nutrients from the soil. However, other factors may also play a role, such as a different initial soil composition. Another difference between the practices of the two farmers whose fields were sampled was the fertilization scheme. The farmer that intercropped applied her manure as a topdressing, while the other farmer ploughed the manure into the soil. This too could have led to a difference in the total amount and availability of nutrients in the soil. Further research would need to be done to determine which of the two practices has the greater impact on soil nutrients: intercropping or fertilization method.

Nevertheless, even allowing for such other factors, the absence of any sign of increased nutrient content in the intercropped fields is remarkable, given that growing cowpeas and ploughing them into the soil should drastically increase soil N content. An explanation for this absence of an observed effect is that the intercropping farmer sold the cowpeas on the market rather than ploughing them into the soil. It is likely that the farmer originally took up intercropping to increase soil N after the advice from the visiting extension workers. However, she at some point changed her practice to make use of the opportunity that harvesting the cowpeas offered her in terms of additional income. It is unclear when or why exactly she changed her practice, but it is likely that this was financially motivated. Due to the originally rich soil in the Nyalenda Urban gardens, the farmer also saw no noticeable reduction in yield as a consequence.

The interviews and FGDs with the Mesopotamia group showed that there is knowledge present of a wide range of agricultural management practices. However, the interviews with the individual women members of the group showed that the knowledge on these practices is unequally distributed and that while they may be known to a technique they do not possess technical knowledge on the effects of their management practices. We conclude that the incomplete knowledge of these farmers is a consequence of the way they acquire and rank knowledge, as well as their lack of access to alternative sources such as the internet. Further research would be needed to confirm this conclusion.

While this paper covers a case study of limited scale, meaning that this should be taken into consideration when viewing the results and drawing conclusions, the circumstances found within the Mesopotamia group are representative for many other groups in the urban gardens of cities in Kenya and subtropical Africa. The case study showed that women's decisions in agricultural management are influenced by their socio-economic and cultural status. Trainings should be adapted to take the socio-economic circumstances of the trainees into account. Furthermore, the gender differences in ability and access should similarly be taken into account in order to improve the effectiveness of a given training or agricultural recommendations.

**Data Availability**

Additional soil chemical data is available can be found in the supplements of this article. Any additional data from the interviews or focus group discussions can be requested through the corresponding author.

## Author Contributions

NTJ suggested the study and contributed to the data analysis. EK contributed the soil data and the interview and focus group discussion data. KK, BJ, and NP contributed to the data analysis and background data on soil and social sciences.

## Competing interests

The authors declare that there is no conflict of interest.

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
