# Peer review of "Women's agricultural practices and their effects on soil nutrient content in the Nyalenda urban gardens of Kisumu, Kenya"

_SOIL, 2018_

## Referee Comment (RC1) · Msc Hemminger (Referee) · 29 Aug 2018

General Comment: The case study combines in an innovative way soil nutrient analysis with farmer interviews. This approach is very useful n order to derive management recommendations that are feasible to the farmers. However the research questions should be formulated more clearly and it should be explained how they were developed from existing literature. Being a case study, it is important to explain which general conclusions can be made from the results.

Specific Comments:

[Figure]

1. Does the paper address relevant scientific questions within the scope of SOIL? yes

2. Does the paper present novel concepts, ideas, tools, or data? The data on soil nutrient content are new.

3. Does the paper address soils within a multidisciplinary context? yes

4. Is the paper of broad international interest? The relevance and relation to results and questions of international research still needs to be better explained. There is a growing body of research on urban agriculture in Africa, which is not sufficiently mentioned (see e.g. Orsini et al. 2013, Hamilton et al. 2014 –>please see the reference list in the supplement). Regarding Gender Analysis it would be interesting to analyse whether the plots managed by women have a different soil nutrient status than those of men (see literature on Gender Gap in agricultural Productivity) and what constraints women face in their production (access to resources and time issues, ("triple burden" childcare, production and community tasks)

5. Are clear objectives and/or hypotheses put forward? I think your question "how does women's knowledge influence soil nutrient content through their management" is not quite clear. Do you propose the hypothesis that higher knowledge will lead women to apply more effective management practices and the soil nutrient content will be higher? Consider that knowledge of a technique does not equal implementation of the technique. There might be financial or time constraints and also cultural and individual factors that influence a person's decision to use a certain agricultural practice. Your results show that an advocated technique (intercropping) leads to lower soil nutrient content, did you propose that the women using this technique had less or more knowledge? Maybe it could be an idea to structure your objectives like this: Aim: Derive recommendations for soil management in urban gardens in Kisumu, Kenya Questions: a) What is the soil nutrient content? (Discuss whether the results you are found are favourable or nonfavourable for agricultural production, should the nutrient content be raised? Might leaching be a problem etc.) b) Which of the recommended soil management practices (suggested based on evidence of agricultural science) are feasible to the women farmers? c) What are research gaps and limits of current agricultural extension activities?

6. Are the scientific methods valid and clear outlined to be reproduced? There is still information missing: What method did you use to choose the sample plots? In how far are they representative for the area? Regarding the interview results, there is not sufficiently stated which information was gained from the 2 women farmers cultivating the sample plots, the women group and the mixed group. Did the two women farmers cultivating the plots participate in the FGD? Why did you choose to organize a female and a mixed group instead of a female and a male group, which would have allowed for comparison of male and female knowledge?

7. Is the soil type/classification adequately described? In your abstract and introduction you refer to nutrient deficiencies in Kenyan agricultural soils and poor soil management as one possible cause. Yet your results are that soil nutrient content is high for both sample plots. Did you record the amount of fertilizer and organic material that was applied to the fields by the farmers? Are the plots examples of high input vegetable production and thus difficult to compare to the average (rural) agricultural soils? (see Predotova et al. 2011; Lompo et al. 2012). Is the overall decline in agricultural productivity in Kenya also observed in Urban agriculture?

8. Are analyses and assumptions valid? See above

9. Are the presented results sufficient to support the interpretations and associated discussion? I think the presentation of the soil nutrient analysis is clear. Please try to document the interview results more clearly. What are interview results, what are FGD results? E.g. how many of the participants know that plants need nutrients from the soil? With which questions did you measure technical knowledge?...

10. Is the discussion relevant and backed up? Be careful not to mention new results in the discussion part (page 9, 6-15) and do not discuss your results in the results section

(p.12 12-14).

11. Are accurate conclusions reached based on the presented results and discussion? I think the conclusion is written very clearly, could you add your conclusion whether intercropping is useful or not? When you mention gender-differentiated knowledge, could you specify in your results what knowledge was specifically male or female? Did men have less sensory knowledge than women? Did men have more technical knowledge than women? What could be advantages of the traditional practical and sensory knowledge these women have? Do you have results whether male and female farmers apply different techniques and have different yields?

12. Do the authors give proper credit to related and relevant work and clearly indicate their own original contribution? You clearly indicated your own contribution. Please have a look at the FAO State of Food and Agriculture Report 2010-2011 "Women in Agriculture- Closing the gender gap for development" and Doss et al. 2018 regarding women having lower yields than men in dev. countries (p. 3, l 15)

13. Does the title clearly reflect the contents of the paper and is it informative? For me nutrient content in relation to knowledge is not clear (see point 5 above)

14. Does the abstract provide a concise and complete summary, including quantitative results? The introduction part in the abstract could be shorter and should mention urban agriculture.

15. Is the overall presentation well structured? I think starting the introduction with the global relevance of your topic would help to understand your research aim.

16. Is the paper written concisely and to the point? yes

17. Is the language fluent, precise, and grammatically correct? yes

18. Are the figures and tables useful and all necessary? yes

19. Are mathematical formulae, symbols, abbreviations, and units correctly defined
and used according to the author guidelines?

20. Should any parts of the paper (text, formulae, figures, tables) be clarified, reduced, combined, or eliminated? Please clarify the legend of figures 1a-f, available, exchangeable and total (Does total include available and exchangeable?, then the color scheme is misleading)

21. Are the number and quality of references appropriate? Please see the references below.

22. Is the amount and quality of supplementary material appropriate and of added value? yes

References

Doss, C., Meinzen-Dick, R., Quisumbing, A., Theis, S. Women in agriculture: Four myths (2018) Global Food Security, 16, pp. 69-74. DOI: 10.1016/j.gfs.2017.10.001

Hamilton, A.J., Burry, K., Mok, H.-F., Barker, S.F., Grove, J.R., Williamson, V.G. Give peas a chance? Urban agriculture in developing countries. A review (2014) Agronomy for Sustainable Development, 34 (1), pp. 45-73. DOI: 10.1007/s13593-013-0155-8

Hovorka, A.J. Urban agriculture: Addressing practical and strategic gender needs (2006) Development in Practice, 16 (1), pp. 51-61. Cited 27 times. DOI: 10.1080/09614520500450826

Lompo, D.J.-P., Sangaré, S.A.K., Compaoré, E., Papoada Sedogo, M., Predotova, M., Schlecht, E., Buerkert, A. Gaseous emissions of nitrogen and carbon from urban vegetable gardens in bobo-dioulasso, burkina Faso(2012) Journal of Plant Nutrition and Soil Science, 175 (6), pp. 846-853. DOI: 10.1002/jpln.201200012

Orsini, F., Kahane, R., Nono-Womdim, R., Gianquinto, G. Urban agriculture in the developing world: A review (2013) Agronomy for Sustainable Development, 33 (4), pp. 695-720. Cited 86 times. DOI: 10.1007/s13593-013-0143-z

Predotova, M., Bischoff, W.-A., Buerkert, A. Mineral-nitrogen and phosphorus leaching from vegetable gardens in Niamey, Niger (2011) Journal of Plant Nutrition and Soil Science, 174 (1), pp. 47-55. DOI: 10.1002/jpln.200900255

Theis, S., Lefore, N., Meinzen-Dick, R., Bryan, E. What happens after technology adoption? Gendered aspects of small-scale irrigation technologies in Ethiopia, Ghana, and Tanzania (2018) Agriculture and Human Values, 35 (3), pp. 671-684. Cited 1 time. DOI: 10.1007/s10460-018-9862-8

———————————————————————

---

## Referee Comment (RC2) · Anonymous Referee #2 · 21 Sep 2018

General Comment: The paper aims at combining soil nutrient analysis with women's agricultural knowledge and their management decisions. While in general this is an important question, the paper is lacking theoretical and empirical (data) depth.

Specific Comments:

1. Does the paper address relevant scientific questions within the scope of SOIL? yes

2. Does the paper present novel concepts, ideas, tools, or data? New data, but too little to be of real relevance.

3. Does the paper address soils within a multidisciplinary context? Yes

4. Is the paper of broad international interest? Theoretically yes, this paper could be of interest. In practice, however the data are too limited in scope and the outlined research question is not really thoroughly addressed (one option might be to reformulate the research question, depending on the data that is available)

5. Are clear objectives and/or hypotheses put forward? While a clear objective is set "understanding how women's knowledge influences soil management and thereby the soil nutrient status", it is not clearly answered. E.g. has any effort been put into understanding whether intercropping or not is influenced by knowledge? Or what the role of knowledge is in the decision to plough manure into the soil, or not?

6. Are the scientific methods valid and clear outlined to be reproduced? The methods as such seem to be okay, but the data presented is insufficient. Information of the history of soil is missing (how long have they been cultivated with the different method). . .., quantitative estimation about the amount of manure applied is also missing, Sampling on only four fields is not really representative. . . It is unclear how the sampling plots have been chosen. . . ... The interview results should be presented in more depth.

9. Are the presented results sufficient to support the interpretations and associated discussion?

I would say the presented results are sometimes unclear or even contradictory. e.g. 5 the paper states that people have limited technical knowledge just to continue a few lines letter saying that the "women spoke of a variety of agricultural meetings". The difference to the knowledge of men is not made clear. In general the difference between male and female knowledge should be made clear. And it should also be shown how the techniques of men and women differ because of differences in knowledge.

Another example: Some statements like "no fallow periods because of a lack of land" could be analysed more deeply in order to understand how knowledge is influencing this statement.

[Figure]

11. Are accurate conclusions reached based on the presented results and discussion? From what I can see the main difference in the soils might come from a higher SOM on the plots where no intercropping is made (SOM as important for CEC). The interesting question would however by, why there is more manure on the plots without intercropping. This might help to understand the reasons behind the different outcomes more clearly. Related to this it could be discussed, whether people should know about the difference (in case the difference is influenced by management practices).

15. Is the overall presentation well structured? The paper is well structured. However the introduction is not really introducing the state of the art with regards to (female) soil knowledge and management practices. ... The general truths for overall agriculture in Kenya, might be good to justify the research, however they are not really relevant in answering the question and are a bit too general.

---

## Author Comment (AC1) · 27 Sep 2018

Dear Referee, we would like to thank you for taking the time to read this paper and writing your review. Based on your feedback and that of the other reviewer we hope to revise our manuscript. With this reply we hope to address your specific concerns and comments.

General Comment: The case study combines in an innovative way soil nutrient analysis with farmer interviews. This approach is very useful in order to derive management recommendations that are feasible to the farmers. However the research questions should be formulated more clearly and it should be explained how they were developed

from existing literature. Being a case study, it is important to explain which general conclusions can be made from the results.

Reply: More than creating management recommendations, this case study is meant to create insight among scientists and policy makers alike, and show that when recommendations are made they must be tailored to more than the soil/environment – the receiver and their socio-economic situation are equally if not more important. We received similar feedback from the other reviewer regarding our research questions and we realize that we may have formulated the main research question too broadly for the scope of the research. We will reformulate the research question based on the advice given. We wish to emphasize that, while indeed a case study, the sites and setup were carefully selected in close partnership with the local stakeholders (research organizations, NGO's and the women groups themselves) to ensure a representative case study for a phenomenon that is wide spread throughout the developing world. We realize that we did not explain this well enough, and will elaborate the selection process in a revised version.

Comment 4. Is the paper of broad international interest? The relevance and relation to results and questions of international research still needs to be better explained. There is a growing body of research on urban agriculture in Africa, which is not sufficiently mentioned (see e.g. Orsini et al. 2013, Hamilton et al. 2014 –>please see the reference list in the supplement). Regarding Gender Analysis it would be interesting to analyse whether the plots managed by women have a different soil nutrient status than those of men (see literature on Gender Gap in agricultural Productivity) and what constraints women face in their production (access to resources and time issues, ("triple burden" childcare, production and community tasks)

Reply: The direct comparison of men and women was not within the scope of this study. Instead we focused on analyzing the interplay of soil fertility, management practices, and knowledge transfer and decision making processes in urban farming women food entrepreneur groups. The rationale to focus on women food entrepreneur groups

is that they are very important in the developing world, yet chronically understudied scientifically and insufficiently recognized by societal stakeholders such as policy makers and NGOs. Though we have tried to use international research to show the relevance and relation of our case study in the broader context we may not have been entirely successful in achieving this. We would like to thank the reviewer for providing us a list of interesting references that we will certainly use to expand and strengthen our manuscript in our revisions if we are given the opportunity.

Comment 5. Are clear objectives and/or hypotheses put forward? I think your question "how does women's knowledge influence soil nutrient content through their management" is not quite clear. Do you propose the hypothesis that higher knowledge will lead women to apply more effective management practices and the soil nutrient content will be higher? Consider that knowledge of a technique does not equal implementation of the technique. There might be financial or time constraints and also cultural and individual factors that influence a person's decision to use a certain agricultural practice. Your results show that an advocated technique (intercropping) leads to lower soil nutrient content, did you propose that the women using this technique had less or more knowledge?

Reply: We found that the choice of whether to apply the intercropping technique was actually not based on knowledge, but rather that there was a socio-economic motivation as you also suggest. We may have failed to convey this well enough in the present version of our manuscript and will correct this in our revisions. We found that women's knowledge does impact their agricultural management practices, which in turn influences their soil's nutrient content – however their main motivation for choosing one management practice over another was based on personal circumstances. The women practicing intercropping had incomplete knowledge regarding the technique, leading them to improperly apply it, however this improper application led to an improvement of their finances by yielding an additional crop to sell, which gave incentive to continue. The soil in this urban garden is of sufficient quality and fertility so

that there is no noticeable difference in crop quality for the women regardless of their chosen management technique. Again, we will more clearly explain this in a revised version.

-Continued Comment 5- Maybe it could be an idea to structure your objectives like this: Aim: Derive recommendations for soil management in urban gardens in Kisumu, Kenya Questions: a) What is the soil nutrient content? (Discuss whether the results you are found are favourable or nonfavourable for agricultural production, should the nutrient content be raised? Might leaching be a problem etc.) b) Which of the recommended soil management practices (suggested based on evidence of agricultural science) are feasible to the women farmers? c) What are research gaps and limits of current agricultural extension activities?

Reply: The suggested restructuring, while very interesting, falls beyond the scope of this case study. Such a comparison would constitute an entire new research project in its own right. As explained before, it was not our aim to create recommendations for soil management – rather we sought to understand the motivation and the knowledge base of the women farmers of the urban gardens and their impact on the soils nutrient status. The feasibility of other management practices were not included in this study and as the reviewer suggests above and is shown by this research such recommendations cannot be done based purely on soil science/agricultural research if you would like to see implementation.

However, based on our research we can formulate recommendations with respect to the current management practices for the groups involved in the present case study. The nutrient content in their urban gardens is sufficient considering the current practices, crops grown and management schemes. The farmers do not use mineral/artificial fertilizers and leaching is a minimal problem. Therefore, we can conclude that the current practices are sustainable and there is no pressing need to change to maintain the status quo. With respect to intercropping, correct application (as opposed to the incorrect application performed by the groups now) would not be expected to lead to

higher crop yields, as N contents are already sufficient for the crops grown. As per the reviewers suggestion, we will include a more detailed reflection along these lines in a revised version.

Comment 6. Are the scientific methods valid and clear outlined to be reproduced? There is still information missing: What method did you use to choose the sample plots? In how far are they representative for the area? Regarding the interview results, there is not sufficiently stated which information was gained from the 2 women farmers cultivating the sample plots, the women group and the mixed group. Did the two women farmers cultivating the plots participate in the FGD? Why did you choose to organize a female and a mixed group instead of a female and a male group, which would have allowed for comparison of male and female knowledge?

Reply: This paper describes a case study that has been carefully setup with the local partners from scientific institutes and NGOs with a vast experience in the area, and indeed with the women farmer groups themselves. This in itself is in our view a unique approach that, by extensive use of the local expertise, ensures the case study is representative of a typical urban gardening situation that can be found abundantly throughout Kenya, sub-Saharan Africa and indeed the developing world. We realize we may not have explained the selection process and representability of the case study well enough and aim to do this if we are given the opportunity to revise.

It is clear, based on both this review and the 2ndÂň, that we have not been diligent enough in describing the results of the interviews and the focus group discussions – the raw data of which was not included in the dataset for privacy reasons. The two women farmers whose fields were used in the soil analysis also participated in the interviews and focus group discussions – we look to make more use of this data in the revisions if given the opportunity. It was beyond the scope of this case study to directly compare male and female farmers, which is why there were no separate meetings with only male farmers.

Comment 7. Is the soil type/classification adequately described? In your abstract and introduction you refer to nutrient deficiencies in Kenyan agricultural soils and poor soil management as one possible cause. Yet your results are that soil nutrient content is high for both sample plots. Did you record the amount of fertilizer and organic material that was applied to the fields by the farmers? Are the plots examples of high input vegetable production and thus difficult to compare to the average (rural) agricultural soils? (see Predotova et al. 2011; Lompo et al. 2012). Is the overall decline in agricultural productivity in Kenya also observed in Urban agriculture?

Comment 8. Are analyses and assumptions valid? See above

Reply: The nutrient content is high in the recorded urban garden likely due to the richness of the soil's parent material, not necessarily due to the farmers application of manure. Exact amounts of manure/compost were not noted, but an inventory was made of fertilizers used, as well as fertilization methods and frequency. The case study is meant to be representative of a typical urban gardening situation and as such can't be directly compared to the rural soils. Production and demand for vegetables from urban gardens are high, but the soils and crops are very different from rural areas. There is no evidence of decline in productivity, rather the opposite – this is one of the ways in which urban gardening differs from traditional (rural) agriculture.

Comment 9. Are the presented results sufficient to support the interpretations and associated discussion? I think the presentation of the soil nutrient analysis is clear. Please try to document the interview results more clearly. What are interview results, what are FGD results? E.g. how many of the participants know that plants need nutrients from the soil? With which questions did you measure technical knowledge?...

Reply: As noted at point 6, we hope to be allowed to more fully incorporate the interviews and focus group discussion results in our revision of the manuscript as this is something that was noted as missing by both this reviewer and the 2nd reviewer. Only a selection of the acquired data was included in the paper. The interview data and

the data from the focus group discussion was more extensive than shown and was not included in the attached dataset for privacy related reasons. In revisions we will more fully incorporate these results, whilst continuing to respect privacy.

Comment 10. Is the discussion relevant and backed up? Be careful not to mention new results in the discussion part (page 9, 6-15) and do not discuss your results in the results section (p.12 12-14).

Reply: We would like to thank the reviewer for pointing out these instances, we shall correct them.

Comment 11. Are accurate conclusions reached based on the presented results and discussion? I think the conclusion is written very clearly, could you add your conclusion whether intercropping is useful or not? When you mention gender-differentiated knowledge, could you specify in your results what knowledge was specifically male or female? Did men have less sensory knowledge than women? Did men have more technical knowledge than women? What could be advantages of the traditional practical and sensory knowledge these women have? Do you have results whether male and female farmers apply different techniques and have different yields?

Reply: The usefulness of intercropping is largely dependent on ones goals and techniques. As the women are currently doing the intercropping does not improve their soil quality, however it does have positive effect on their financial situation. Considering the richness of their soil there is no direct reason to discourage these farmers from their current practices. Unfortunately it is not possible for us to show the direct differences between the soil nutrient condition of the men's fields and the women's fields or their technical knowledge as no separate male fields were tested, nor were they included in the interviews. This was simply beyond the scope of this case study – though objectively it would be very interesting to see if such differences could be found. While we lack sufficient data to include the roles of men in the paper, we have noted that men tend to have more access to capital and means, meaning that their practices often differ from that of women on that basis and because of this they also have a different view of agricultural problems.

Comment 12. Do the authors give proper credit to related and relevant work and clearly indicate their own original contribution? You clearly indicated your own contribution. Please have a look at the FAO State of Food and Agriculture Report 2010-2011 "Women in Agriculture- Closing the gender gap for development" and Doss et al. 2018 regarding women having lower yields than men in dev. countries (p. 3, l 15)

Reply: We did note from literature that women generally have lower yields than men in developing countries, but that this often has to do with a lack of access and means on the side of the women. We will note your reference and see to include it in our revisions.

Comment 13. Does the title clearly reflect the contents of the paper and is it informative? For me nutrient content in relation to knowledge is not clear (see point 5 above)

Reply: As stated at point 5, the knowledge of the women does not directly influence their choice of management practice, rather their socio-economic situation generally does. The knowledge of the women does however impact the way they implement their chosen management practice, which in turn influences the soil nutrient content. We hope we will be able to convey this more clearly if given the opportunity to make revisions to the manuscript.

Comment 14. Does the abstract provide a concise and complete summary, including quantitative results? The introduction part in the abstract could be shorter and should mention urban agriculture.

Reply: A valid point, we will include that during our revision of the manuscript.

Comment 15. Is the overall presentation well structured? I think starting the introduction with the global relevance of your topic would help to understand your research aim.

[Figure]

Reply: This is a difficult point as our other reviewer notes that they consider our introduction a bit too broad, we will have to consider how to more carefully balance this in our revisions.

Comment 20. Should any parts of the paper (text, formulae, figures, tables) be clarified, reduced, combined, or eliminated? Please clarify the legend of figures 1a-f, available, exchangeable and total (Does total include available and exchangeable?, then the color scheme is misleading).

Reply: We would like to thank the reviewer for pointing this out and will attempt to make the figure more clear in the revisions. The total does in fact include the available and exchangeable.

Comment 21. Are the number and quality of references appropriate? Please see the references below.

Reply: Overall we would like to again thank the review for their valuable comments and also their included list of refer

---

## Author Comment (AC2) · 27 Sep 2018

Dear Referee, first of all we would like to thank you for taking the time to read this paper and writing your review. We hope to edit the manuscript to address your concerns and would like to address your specific feedback and comments with this reply.

General Comment: The paper aims at combining soil nutrient analysis with women's agricultural knowledge and their management decisions. While in general this is an important question, the paper is lacking theoretical and empirical (data) depth.

Specific Comments: 2. Does the paper present novel concepts, ideas, tools, or data?

[Figure]

New data, but too little to be of real relevance.

Reply: It is true that the paper presents the results of a case study. However, it is a case study that has been carefully setup with the local partners from scientific institutes and NGOs with a vast experience in the area, and indeed with the women farmer groups themselves. This in itself is in our view a unique approach that, by extensive use of the local expertise, ensures the case study is representative of a typical urban gardening situation that can be found abundantly throughout Kenya, sub-Saharan Africa and indeed the developing world. We realize we may not have explained the selection process and representability of the case study well enough and aim to do this if we are given the opportunity to revise.

With respect to the perceived data paucity, it should be noted that only a selection of the acquired data was included in the paper. The interview data and the data from the focus group discussion was more extensive than shown and was not included in the attached dataset for privacy related reasons. In revisions we will more fully incorporate these results, whilst continuing to respect privacy.

Comment 4. Is the paper of broad international interest? Theoretically yes, this paper could be of interest. In practice, however the data are too limited in scope and the outlined research question is not really thoroughly addressed (one option might be to reformulate the research question, depending on the data that is available)

Reply: As noted above, the study was carefully selected as a representative case study for a phenomenon that is wide spread throughout the developing world. In addition, we will look to incorporate the data from the interviews and focus group discussions more fully to support our conclusions with revisions. At the same time we realize that we may have formulated the main research question too broadly for the scope of the research. We are confident that revising the results from the social sciences section as well as re-examining our research question will allow us to address these concerns.

Comment 5. Are clear objectives and/or hypotheses put forward? While a clear objective is set "understanding how women's knowledge influences soil management and thereby the soil nutrient status", it is not clearly answered. E.g. has any effort been put into understanding whether intercropping or not is influenced by knowledge? Or what the role of knowledge is in the decision to plough manure into the soil, or not?

Reply: It is clear that we did not formulate our conclusions well enough and we would like to thank the reviewer for pointing this out. For example, we had hoped to convey that intercropping as done by the women farmers in Nyalenda was imperfectly done due to gaps in the knowledge. While the women farmers have the basics of this management practices, i.e. the intercropping with a legume to improve soil N, they lack the technical knowledge to properly apply this practices. These women do not plough the legume into the soil after a certain period of time to maximize soil N input, but rather let the legume grow to maturity to harvest it as crop. This essentially leads to a more rapid extraction of nutrients from the soil. The agricultural meetings the women attend are useful, but knowledge transfer there is incomplete because of several socio-economic barriers. We will formulate this more clearly and extensively in a revised version, and see that it is better supported by data from the interviews and focus group discussions.

Comment 6. Are the scientific methods valid and clear outlined to be reproduced? The methods as such seem to be okay, but the data presented is insufficient. Information of the history of soil is missing (how long have they been cultivated with the different method): : :., quantitative estimation about the amount of manure applied is also missing, Sampling on only four fields is not really representative: : : It is unclear how the sampling plots have been chosen: : :.. The interview results should be presented in more depth.

Reply: While we do not have all the background information mentioned in the 6th comment, such as quantitative estimates of manure use; there is more information regarding the history of the site available than incorporated in the article, both from literature sources and the interviews and focus groups discussions. The limited amount of fields used in the study is both a reflection of the limitations of this study as a case

study, as well as an attempt to make the influence of the two management practices as comparable as possible. The four fields were chosen, in careful consultation with our local partners including the women groups themselves, for being most representative for the studied management practices. The interview results as noted before can and will presented more in depth with revision of the manuscript.

Comment 9. Are the presented results sufficient to support the interpretations and associated discussion? I would say the presented results are sometimes unclear or even contradictory. e.g. 5 the paper states that people have limited technical knowledge just to continue a few lines letter saying that the "women spoke of a variety of agricultural meetings". The difference to the knowledge of men is not made clear. In general the difference between male and female knowledge should be made clear. And it should also be shown how the techniques of men and women differ because of differences in knowledge. Another example: Some statements like "no fallow periods because of a lack of land" could be analysed more deeply in order to understand how knowledge is influencing this statement.

Reply: We would like to thank the reviewer for pointing out that the results are sometimes unclear or seem contradictory and will seek to clarify the results where necessary. Likely a more thorough incorporation of the interview and focus group discussions will address the main concerns. For the specific cases mentioned, while the women spoke of a variety of agricultural management practices during the meetings and interviews they lack the technical knowledge – meaning that they have heard or tried techniques, but did often lack complete knowledge regarding their proper application. An example being that those who practice intercropping did not realize that they had to plough the entire plant into the soil about 3 weeks after planting for the soil to benefit from the intercropping.

The knowledge and techniques of men are not explored further in the paper as they were not the focus of the research and they were only included in the focus group discussions. The choice to focus on women is based on the fact that women very often

play a leading role in the urban gardening practices, yet their socio-economic position as well as the dynamics of their contribution forms a seriously understudied area. As indicated before, it is clear that we failed to explain the selection of our sites and focus well enough and aim to amend this in a revised version. While we lack sufficient data to include the roles of men in the paper, we have noted that men tend to have more access to capital and means, meaning that their practices often differ from that of women on that basis and because of this they also have a different view of agricultural problems.

The statement regarding the lack of fallow periods could perhaps be further expanded with results from the interviews – the lack of fallow periods is not because the women farmers lack the knowledge regarding this practice, rather it is their need for revenue forcing them to continue using the land In spite of their awareness that they ought to rest the land.

Comment 11. Are accurate conclusions reached based on the presented results and discussion? From what I can see the main difference in the soils might come from a higher SOM on the plots where no intercropping is made (SOM as important for CEC). The interesting question would however by, why there is more manure on the plots without intercropping. This might help to understand the reasons behind the different outcomes more clearly. Related to this it could be discussed, whether people should know about the difference (in case the difference is influenced by management practices).

Reply: The reviewer raises a valid point and while we lack quantative data regarding the use of manure – which may of course be of a large influence on SOM, we do know that at the very least the application method differs. The application method could have a similar if not just an important effect on the SOM levels of the soil as the amount of manure – we could expand on this further during revision of the manuscript.

Comment 15. Is the overall presentation well structured? The paper is well structured. However the introduction is not really introducing the state of the art with regards to

(female) soil knowledge and management practices: : :. The general truths for overall agriculture in Kenya, might be good to justify the research, however they are not really relevant in answering the question and are a bit too general.

Reply: The shortcomings of the introduction and background were also mentioned by the other reviewer and we are thankful for both of them for pointing this out. We will seek to improve this when allowed to revise, by using more and also more up to date literature.

Overall we would like to again thank the review for their valuable comments and we will look to use their feedback in the editing of our manuscript.

---

## Author Response (AR1)

**Contents:**

**Review 1 & Response:**

Dear Referee, we would like to thank you for taking the time to read this paper and writing your review. Based on your feedback and that of the other reviewer we hope to revise our manuscript. With this brief we hope to address your specific concerns and comments.

*General Comment: The case study combines in an innovative way soil nutrient analysis with farmer interviews. This approach is very useful in order to derive management recommendations that are feasible to the farmers. However the research questions should be formulated more clearly and it should be explained how they were developed from existing literature. Being a case study, it is important to explain which general conclusions can be*
10  *made from the results.*

More than creating management recommendations, this case study is meant to create insight among scientists and policy makers and show that when recommendations are made they must be tailored to more than the soil/environment – the receiver and their socio-economic situation are equally if not more important. We received similar feedback from the other
15  reviewer regarding our research questions and we realize that we may have formulated the main research question too broadly for the scope of the research and will look to see if we can more carefully rephrase it during revisions if given the chance. We maintain that he study was carefully selected as a representative case study for a phenomenon that is wide spread throughout the developing world.

*4. Is the paper of broad international interest? The relevance and relation to results and questions of international*
20  *research still needs to be better explained. There is a growing body of research on urban agriculture in Africa, which is not sufficiently mentioned (see e.g. Orsini et al. 2013, Hamilton et al. 2014 –>please see the reference list in the supplement). Regarding Gender Analysis it would be interesting to analyse whether the plots managed by women have a different soil nutrient status than those of men (see literature on Gender Gap in agricultural Productivity) and what constraints women face in their production (access to resources and time issues, ("triple*
25  *burden" childcare, production and community tasks)*

The direct comparison of men and women was not within the scope of this study and there is also no data available from this study to do such an analysis. Though we have tried to use international research to show the relevance and relation of our case study in the broader context we may not have been entirely successful in achieving this. We would like to thank the
30  reviewer for providing us a list of interesting references that we will certainly explore in our revisions if we are given the opportunity.

*5. Are clear objectives and/or hypotheses put forward? I think your question "how does women's knowledge influence soil nutrient content through their management" is not quite clear. Do you propose the hypothesis that higher knowledge will lead women to apply more effective management practices and the soil nutrient content will be higher? Consider that knowledge of a technique does not equal implementation of the technique. There might be financial or time constraints and also cultural and individual factors that influence a person's decision to use a certain agricultural practice. Your results show that an advocated technique (intercropping) leads to lower soil nutrient content, did you propose that the women using this technique had less or more knowledge?*

We found that the choice of whether to apply the intercropping technique was actually not based on knowledge, but rather that there was a socio-economic motivation as you also suggest. We may have failed to properly formulate this in our results and conclusions and will have to correct this in our revisions. We found that women's knowledge does impact their agricultural management practices, which in turn influences their soil's nutrient content – however their main motivation for choosing one management practice over another was based on personal circumstances. The women practicing intercropping had incomplete knowledge regarding the technique, leading them to improperly apply it, however this improper application led to an improvement of their finances which gave incentive to continue. The soil in this urban garden is of sufficient quality that there is no noticeable difference in crop quality for the women regardless of their chosen management technique.

-Continued- *Maybe it could be an idea to structure your objectives like this: Aim: Derive recommendations for soil management in urban gardens in Kisumu, Kenya Questions: a) What is the soil nutrient content? (Discuss whether the results you are found are favourable or nonfavourable for agricultural production, should the nutrient content be raised? Might leaching be a problem etc.) b) Which of the recommended soil management practices (suggested based on evidence of agricultural science) are feasible to the women farmers? c) What are research gaps and limits of current agricultural extension activities?*

The suggested restructuring is a bit beyond the scope of this case study and another research project all together. We have never attempted to create recommendations for soil management – rather we sought to understand the motivation and the knowledge base of the women farmers of the urban gardens and their impact on the soils nutrient status. The nutrient content in the urban gardens is sufficient considering the current practices and management schemes. The farmers do not use mineral/artificial fertilizers and leaching is a minimal problem. The feasibility of other management practices were not included in this study and as the reviewer suggests above and is shown by this research such recommendations can't be done based purely on soil science/agricultural research if you would like to see implementation.

*6. Are the scientific methods valid and clear outlined to be reproduced? There is still information missing: What method did you use to choose the sample plots? In how far are they representative for the area? Regarding the*

*interview results, there is not sufficiently stated which information was gained from the 2 women farmers cultivating the sample plots, the women group and the mixed group. Did the two women farmers cultivating the plots participate in the FGD? Why did you choose to organize a female and a mixed group instead of a female and a male group, which would have allowed for comparison of male and female knowledge?*

This paper describes a case study that has been carefully setup with the local partners from scientific institutes and NGOs with a vast experience in the area, and indeed with the women farmer groups themselves. This in itself is in our view a unique approach that, by extensive use of the local expertise, ensures the case study is representative of a typical urban gardening situation that can be found abundantly throughout Kenya, sub-Saharan Africa and indeed the developing world.

10    We realize we may not have explained the selection process and representability of the case study well enough and aim to do this if we are given the opportunity to revise.

It is clear, based on both this review and the $2^{nd}$, that we have not been diligent enough in describing the results of the interviews and the focus group discussions – the raw data of which was not included in the dataset for privacy reasons. The two women farmers whose fields were used in the soil analysis also participated in the interviews and focus group

15    discussions – we look to make more use of this data in the revisions if given the opportunity. It was beyond the scope of this case study to directly compare male and female farmers, which is why there were no separate meetings with only male farmers.

*7. Is the soil type/classification adequately described? In your abstract and introduction you refer to nutrient*

20    *deficiencies in Kenyan agricultural soils and poor soil management as one possible cause. Yet your results are that soil nutrient content is high for both sample plots. Did you record the amount of fertilizer and organic material that was applied to the fields by the farmers? Are the plots examples of high input vegetable production and thus difficult to compare to the average (rural) agricultural soils? (see Predotova et al. 2011; Lompo et al. 2012). Is the overall decline in agricultural productivity in Kenya also observed in Urban agriculture?*

*8. Are analyses and assumptions valid? See above*

The nutrient content is high in the recorded urban garden likely due to the richness of the soil's parent material, not necessarily due to the farmers application of manure. Exact amounts of manure/compost were not noted, but an inventory

30    was made of fertilizers used, as well as fertilization methods and frequency. The case study is meant to be representative of a typical urban gardening situation and as such can't be directly compared to the rural soils. Production and demand for vegetables from urban gardens are high, but the soils and crops are very different from rural areas. There is no evidence of decline in productivity, rather the opposite – this is one of the ways in which urban gardening differs from traditional (rural) agriculture.

*9. Are the presented results sufficient to support the interpretations and associated discussion? I think the presentation of the soil nutrient analysis is clear. Please try to document the interview results more clearly. What are interview results, what are FGD results? E.g. how many of the participants know that plants need nutrients from the soil? With which questions did you measure technical knowledge?...*

As noted at point 6, we hope to be allowed to more fully incorporate the interviews and focus group discussion results in our revision of the manuscript as this is something that was noted as missing by both this reviewer and the 2[nd] reviewer. Only a selection of the acquired data was included in the paper. The interview data and the data from the focus group discussion was more extensive than shown and was not included in the attached dataset for privacy related reasons. In revisions we will more fully incorporate these results, whilst continuing to respect privacy.

*10. Is the discussion relevant and backed up? Be careful not to mention new results in the discussion part (page 9, 6-15) and do not discuss your results in the results section (p.12 12-14).*

We would like to thank the reviewer for pointing out these instances, we shall correct them.

*11. Are accurate conclusions reached based on the presented results and discussion? I think the conclusion is written very clearly, could you add your conclusion whether intercropping is useful or not? When you mention gender-differentiated knowledge, could you specify in your results what knowledge was specifically male or female? Did men have less sensory knowledge than women? Did men have more technical knowledge than women? What could be advantages of the traditional practical and sensory knowledge these women have? Do you have results whether male and female farmers apply different techniques and have different yields?*

The usefulness of intercropping is largely dependent on ones goals and techniques. As the women are currently doing the intercropping does not improve their soil quality, however it does have positive effect on their financial situation. Considering the richness of their soil there is no direct reason to discourage these farmers from their current practices.

Unfortunately it is not possible for us to show the direct differences between the soil nutrient condition of the men's fields and the women's fields or their technical knowledge as no separate male fields were tested, nor were they included in the interviews. This was simply beyond the scope of this case study – though objectively it would be very interesting to see if such differences could be found. While we lack sufficient data to include the roles of men in the paper, we have noted that men tend to have more access to capital and means, meaning that their practices often differ from that of women on that basis and because of this they also have a different view of agricultural problems.

*12. Do the authors give proper credit to related and relevant work and clearly indicate their own original contribution? You clearly indicated your own contribution. Please have a look at the FAO State of Food and Agriculture Report 2010-2011 "Women in Agriculture- Closing the gender gap for development" and Doss et al. 2018 regarding women having lower yields than men in dev. countries (p. 3, l 15)*

We did note from literature that women generally have lower yields than men in developing countries, but that this often has to do with a lack of access and means on the side of the women. We will note your reference and see to include it in our revisions.

10 *13. Does the title clearly reflect the contents of the paper and is it informative? For me nutrient content in relation to knowledge is not clear (see point 5 above)*

As stated at point 5, the knowledge of the women does not directly influence their choice of management practice, rather their socio-economic situation generally does. The knowledge of the women does however impact the way they implement

15 their chosen management practice, which in turn influences the soil nutrient content. We hope we will be able to convey this more clearly if given the opportunity to make revisions to the manuscript.

*14. Does the abstract provide a concise and complete summary, including quantitative results? The introduction part in the abstract could be shorter and should mention urban agriculture.*

A valid point, we will include that during our revision of the manuscript.

*15. Is the overall presentation well structured? I think starting the introduction with the global relevance of your topic would help to understand your research aim.*

This is a difficult point as our other reviewer notes that they consider our introduction a bit too broad, we will have to consider how to more carefully balance this in our revisions.

*20. Should any parts of the paper (text, formulae, figures, tables) be clarified, reduced, combined, or eliminated?*

30 *Please clarify the legend of figures 1a-f, available, exchangeable and total (Does total include available and exchangeable?, then the color scheme is misleading).*

We would like to thank the reviewer for pointing this out and will attempt to make the figure more clear in the revisions. The total does in fact include the available and exchangeable.

*21. Are the number and quality of references appropriate? Please see the references*
*below.*

Overall we would like to again thank the review for their valuable comments and also their included list of references and we will look to use their feedback in the editing of our manuscript.

**Review 2 & Response:**

Dear Referee, first of all we would like to thank you for taking the time to read this paper and writing your review. We hope to edit the manuscript to address your concerns and would like to address your specific feedback and comments with this letter.

5 *The paper aims at combining soil nutrient analysis with women's agricultural knowledge and their management decisions. While in general this is an important question, the paper is lacking theoretical and empirical (data) depth.*

*Specific Comments:*

*2. Does the paper present novel concepts, ideas, tools, or data? New data, but too little to be of real relevance.*

The paper does concern a case study with a limited scope and the data included may have been too limited to show its relevance. The interview data and the data from the focus group discussion was more extensive than shown and was not included in the attached dataset for privacy related reasons. In revisions we will more fully incorporate these results. Hopefully this will also serve to address the 2[nd] specific comment of this referee. The case study is meant to serve as an

15 example of the situation of urban gardens that can be found in many cities in sub-Saharan Africa.

*4. Is the paper of broad international interest? Theoretically yes, this paper could be of interest. In practice, however the data are too limited in scope and the outlined research question is not really thoroughly addressed (one option might be to reformulate the research question, depending on the data that is available)*

20 As noted above, we will look to incorporate the data from the interviews and focus group discussions more fully to support our conclusions with revisions. At the same time we realize that we may have formulated the main research question too broadly for the scope of the research. Revising the results from the social sciences section as well as re-examining our research question will hopefully allow us to lay these concerns to rest.

25 *5. Are clear objectives and/or hypotheses put forward? While a clear objective is set "understanding how women's knowledge influences soil management and thereby the soil nutrient status", it is not clearly answered. E.g. has any effort been put into understanding whether intercropping or not is influenced by knowledge? Or what the role of knowledge is in the decision to plough manure into the soil, or not?*

30 Our conclusions may not have been formulated well enough and we would like to thank the reviewer for pointing this out. For example, we had hoped to convey that intercropping as done by the women farmers in Nyalenda was imperfectly done due to gaps in the knowledge. While the women farmers have the basics of this management practices, i.e. the intercropping with a legume to improve soil N, they lack the technical knowledge to properly apply this practices. These women do not

plough the legume into the soil after a certain period of time to maximize soil N input, but rather let the legume grow to maturity to harvest it as crop. This essentially leads to a more rapid extraction of nutrients from the soil. We will attempt to formulate this more clearly in revisions and see that it is better supported by data from the interviews and focus group discussions.

*6. Are the scientific methods valid and clear outlined to be reproduced? The methods as such seem to be okay, but the data presented is insufficient. Information of the history of soil is missing (how long have they been cultivated with the different method): : :., quantitative estimation about the amount of manure applied is also missing, Sampling on only four fields is not really representative: : : It is unclear how the sampling plots have*
10   *been chosen: : :.. The interview results should be presented in more depth.*

While we do not have all the background information mentioned in the 6[th] comment, such as quantitative estimates of manure use; there is more information regarding the history of the site available than incorporated in the article, both from literary sources and the interviews and focus groups discussions. The limited amount of fields used in the study is both a
15   reflection of the limitations of this study as a case study, as well as an attempt to make the influence of the two management practices as comparable as possible. The four fields were chosen for being most representative for the studied management practices. The interview results as noted before can and will presented more in depth with revision of the manuscript.

*9. Are the presented results sufficient to support the interpretations and associated discussion? I would say the*
20   *presented results are sometimes unclear or even contradictory. e.g. 5 the paper states that people have limited technical knowledge just to continue a few lines letter saying that the "women spoke of a variety of agricultural meetings". The difference to the knowledge of men is not made clear. In general the difference between male and female knowledge should be made clear. And it should also be shown how the techniques of men and women differ because of differences in knowledge. Another example: Some statements like "no fallow periods*
25   *because of a lack of land" could be analysed more deeply in order to understand how knowledge is influencing this statement.*

We would like to thank the reviewer for pointing out that the results are sometimes unclear or seems contradictory and will seek to clarify the results where necessary. Likely a more thorough incorporation of the interview and focus group
30   discussions will ease some of the concerns. For the specific cases mentioned, while the women spoke of a variety of agricultural management practices during the meetings and interviews they lack the technical knowledge – meaning that they have heard or tried techniques, but did not always have the knowledge regarding its proper application. An example being that those who practice intercropping didn't realize that they had to plough the entire plant into the soil about 3 weeks after planting for the soil to benefit from the intercropping.

The knowledge and techniques of men are not explored further in the paper as they were not the focus of the research and they were only included in the focus group discussions. A more thorough analysis of these might show a more clear difference between the two groups, but the study can't be expanded beyond that as there is no data. We have noted that men tend to have more access to capital and means, meaning that their practices often differ from that of women on that basis and because of this they also have a different view of agricultural problems.

The statement regarding the lack of fallow periods could perhaps be further expanded with results from the interviews – the lack of fallow periods is not because the women farmers lack the knowledge regarding this practice, rather it is their need for revenue forcing them to continue using the land even though they should rest it and know that they should.

*11. Are accurate conclusions reached based on the presented results and discussion? From what I can see the main difference in the soils might come from a higher SOM on the plots where no intercropping is made (SOM as important for CEC). The interesting question would however by, why there is more manure on the plots without intercropping. This might help to understand the reasons behind the different outcomes more clearly. Related to this it could be discussed, whether people should know about the difference (in case the difference is influenced by management practices).*

The reviewer raises a valid point and while we lack quantative data regarding the use of manure – which may of course be of a large influence on SOM, we do know that at the very least the application method differs. The application method could have a similar if not just an important effect on the SOM levels of the soil as the amount of manure – we could expand on this further during revision of the manuscript.

15. Is the overall presentation well structured? The paper is well structured. However the introduction is not really introducing the state of the art with regards to (female) soil knowledge and management practices: : :. The general truths for overall agriculture in Kenya, might be good to justify the research, however they are not really relevant in answering the question and are a bit too general.

The shortcomings of the introduction and background were also mentioned by the other reviewer and we are thankful for the both of them for pointing this out. We will seek to improve this during coming revisions by using more and also more up to date literature.

Overall we would like to again thank the review for their valuable comments and we will look to use their feedback in the editing of our manuscript.

**List Major Changes:**

Introduction:

Addition of section on the specific challenges of urban agriculture

Removal of the paragraph focussed on Kenyan (rural) agriculture

5 More focus on the position of women and their cultural burdens

Specification of the selection of the field site and refinement of study aims

Methods and materials:

Addition of section specifically on the interviews and focus group discussions

Results

10 Addition of section specifically on the results from the interviews and focus group discussions

In-depth information on tested fields from interviews added

Replacement of figure 1, which one reviewer indicated was unclear

References

9 additional references added to list

**Soil nutrient content in relation to women's agricultural knowledge in the urban gardens of Kisumu, Kenya**

[revised manuscript text omitted]

---

## Referee Report (RR1)

1. Does the paper address relevant scientific questions within the scope of SOIL?

   Yes. It could help for understanding if you address also the specific research question you are addressing already in the introduction: what is the soil type of urban garden soils in the case study area? How do different management practices effect nutrient content and availability of the soils? Why do women farmers chose certain soil management techniques?

2. Does the paper present novel concepts, ideas, tools, or data?

   The combination of soil analysis and qualitative analysis is a novel approach.

3. Does the paper address soils within a multidisciplinary context?

   yes

4. Is the paper of broad international interest?

   yes

5. Are clear objectives and/or hypotheses put forward?

   Yes.

6. Are the scientific methods valid and clear outlined to be reproduced?

   Yes

7. Is the soil type/classification adequately described?
8. Are analyses and assumptions valid?

   yes, yet in two lines of argumentation you could be more specific regarding the interpretation of the soil nutrient data: you write "often a larger portion of the nutrients was water soluble or exchangeable"(p. 10,3) and "some of the nutrients" (p. 11,31), when from the figure it appears that this is only true for Mg and Ca. Also it is not clear from the conclusion which management practice (manure only; manure and intercropping for N enrichment; or manure and intercropping for second income) can be recommended for use by other urban farmers.

   You mention that information on daily activities varies too much, in order to be analysed. Did you check for the total work burden of women (can be calculated from the typical working day)? This could give important insights on which management practices are feasible to women. Also the often quoted FAO report that states women could be 30% more productive neglects that women often have higher work load than men, that restricts them in using certain yield increasing methods.

9. Are the presented results sufficient to support the interpretations and associated discussion? yes
10. Is the discussion relevant and backed up?

Please check the discussion for not introducing new results (e.g. p. 11, l. 1-7; l. 16-22 should be in the results section) Discussion could be shorter and highlight important points, i.e. soil analysis results that (recommended practice) of intercropping led to lower soil nutrient content. Extension failed to see needs of farmers for second income crop.

In presenting the FGD results it is not clear in which FG men were present and whether knowledge differences of men and women could be detected.

11. Are accurate conclusions reached based on the presented results and discussion?

    See above.

12. Do the authors give proper credit to related and relevant work and clearly indicate their own original contribution? Yes

13. Does the title clearly reflect the contents of the paper and is it informative?

    Still I don`t like he direct link of knowledge to nutrient content, maybe: Women`s farming practices and their effect on soil nutrient content in the Nyalenda urban gardens

14. Does the abstract provide a concise and complete summary, including quantitative results? yes
15. Is the overall presentation well structured? yes
16. Is the paper written concisely and to the point? Yes, but please check to prevent iterations of the same lines of arguments.
17. Is the language fluent, precise, and grammatically correct? Yes
18. Are the figures and tables useful and all necessary? Yes, maybe include a table on members of focus group discussions and interview partners?
19. Are mathematical formulae, symbols, abbreviations, and units correctly defined and used according to the author guidelines?
20. Should any parts of the paper (text, formulae, figures, tables) be clarified, reduced, combined, or eliminated? See above
21. Are the number and quality of references appropriate? yes
22. Is the amount and quality of supplementary material appropriate and of added value?

---

## Author Response (AR2)

**Contents**

**Review 1 & Response:**

**First of all, we would like to thank the reviewer for taking the time to give clear and structured feedback. Below we address the feedback point by point where there was need for a response. Responses are bolded.**

1. Does the paper address relevant scientific questions within the scope of SOIL?
   Yes. It could help for understanding if you address also the specific research question you are addressing already in the introduction: what is the soil type of urban garden soils in the case study area? How do different management practices effect nutrient content and availability of the soils? Why do women farmers chose certain soil management techniques?

**We have attempted to incorporate the response to these questions in the earlier version of the paper, but were apparently unsuccessful in doing so. In this version of the paper additional attention has been paid to the way the paper is structured, as also recommended by the 2nd reviewer, and we believe that we are now more clear in addressing the answers to these questions in the text.**

   Is the soil type/classification adequately described?
8. Are analyses and assumptions valid?
   yes, yet in two lines of argumentation you could be more specific regarding the interpretation of the soil nutrient data: you write "often a larger portion of the nutrients was water soluble or exchangeable"(p. 10,3) and "some of the nutrients" (p. 11,31), when from the figure it appears that this is only true for Mg and Ca. Also it is not clear from the conclusion which management practice (manure only; manure and intercropping for N enrichment; or manure and intercropping for second income) can be recommended for use by other urban farmers.

**We have made the mentioned sections of text more specific A generic recommendation is not made in the conclusion because any recommendation would need to be tailor-made to the specific individual circumstances of the farmer in question and their soil.**

   You mention that information on daily activities varies too much, in order to be analysed. Did you check for the total work burden of women (can be calculated from the typical working day)? This could give important insights on which management practices are feasible to women. Also the often quoted FAO report that states women could be 30% more productive neglects that women often have higher work load than men, that restricts them in using certain yield increasing methods.

**This is an interesting technique and we will certainly see if we can use this in the future. However the data collected in this study was not in the right format to apply such a quantification.**

9. Are the presented results sufficient to support the interpretations and associated discussion? yes
10. Is the discussion relevant and backed up?

Please check the discussion for not introducing new results (e.g. p. 11, l. 1-7; l. 16-22 should be in the results section) Discussion could be shorter and highlight important points, i.e. soil analysis results that (recommended practice) of intercropping led to lower soil nutrient content. Extension failed to see needs of farmers for second income crop.

**Thank you for pointing these instances out. We have moved some sections of the text and removed some in order to streamline and better structure the text of the paper.**

In presenting the FGD results it is not clear in which FG men were present and whether knowledge differences of men and women could be detected.

**The focus group discussion section of the paper has also been restructured. We believe it is now easier to make the distinction between data from the mixed FGD and the women only FGD.**

11. Are accurate conclusions reached based on the presented results and discussion?
    See above.

12. Do the authors give proper credit to related and relevant work and clearly indicate their own original contribution? Yes

13. Does the title clearly reflect the contents of the paper and is it informative?
    Still I don't like he direct link of knowledge to nutrient content, maybe: Women's farming practices and their effect on soil nutrient content in the Nyalenda urban gardens

**Due to the complex relationships featured in this paper it has been difficult finding a title that covers it in entirety. We have altered the title based on your recommendation and believe that it now infers less of a direct relationship between women's knowledge and soil nutrient content. Thank you for your suggestion.**

14. Does the abstract provide a concise and complete summary, including quantitative results?
    yes
15. Is the overall presentation well structured? yes
16. Is the paper written concisely and to the point? Yes, but please check to prevent iterations of the same lines of arguments.

**There were several instances were information was repeated in the text. In this version these instances have been removed as much as possible.**

17. Is the language fluent, precise, and grammatically correct? Yes
18. Are the figures and tables useful and all necessary? Yes, maybe include a table on members of focus group discussions and interview partners?
19.

**We have included an anonymized version of the participants lists for the interviews and the FGDs in the supplement of this revision.**

20. Are mathematical formulae, symbols, abbreviations, and units correctly defined and used according to the author guidelines?
21. Should any parts of the paper (text, formulae, figures, tables) be clarified, reduced, combined, or eliminated? See above
22. Are the number and quality of references appropriate? yes
23. Is the amount and quality of supplementary material appropriate and of added value?

**We would like to thank the reviewer again for giving clear and structured feedback.**

**Review 2 & Response:**

**First of all we would like to thank the reviewer for going through the paper and providing clear and structured feedback. We address some of the points raised below. Responses are bolded.**

Suggestions for revision or reasons for rejection (will be published if the paper is accepted for final publication)
The aim of this paper is to link women's knowledge on agricultural practices and their motivations in choosing specific practices with the nutrient content of their soils.
The data presented however are too limited in scope to really thoroughly address this question. The paper nevertheless can provide some insights in the way urban agriculture in Nyalenda is practiced and which constraints people face. However, I would suggest some extensive re-structuring (see below) and would furthermore suggest to the author to be more open about the short-comings of their findings right from the beginning: be more open about what they can really "prove" and which open questions remain. The conclusion in this sense would profit from finishing with suggestions for further research that needs to be done.

**We have restructured the different sections of the paper based on your recommendations and those of the other reviewers. We have also tried to make clear more carefully the limitations of the study as you suggest.**

Some specific comments:
Paragraph: p.2, 11-24 not really necessary
p.3, 6-15 – Are you referring to agriculture in general or to urban agriculture?
p.3., 16-22: Which factors did you use to select Nyalenda? In which way is it representative?
p. 3, 27-32, paragraph would rather fit in the intro

Is the Question: how knowledge influences action addressed?
p.6, 6-7: do you know for the two fields whether the farmers sued compost or cow manure? And whether they are the ones that are occasionally using mineral fertilizer?
Compare and combine: p.6, 13-18 and p.7, 30-45: some many similarities in those parts. Best to put all information on p.6
p.7, 34 "know" is missing)

**We have clarified where you indicated that the text caused confusion and incorporated a more detailed account of the selection criteria and the management practices used in the sampled fields. Overlap in the text has been removed as much as possible.**

IN general for the interviews and FGD: differentiate clearly where you could identify a lack of know, where are lack of motivation and where a lack of resources as the reason for a specific management technique and structure the presentation of your findings this way. This should help to get clearer information concerning the aim of your paper: "women knowledge on agricultural practices and their motivations in choosing specific practices with the nutrient content of their soils." Chapter 3.3. in general is a mixture of an interpretation of the soil data, with some interviews and additional information. Some more structure would do good here. A, engage in depth with the soil analysis and which kind of conclusion you can or cannot draw from them and B, Discuss (in the Discussion part?) whether the differences can be explained through the different management techniques.

**We have extensively restructured the paper and believe that the data is presented in a clearer manner in this version.**

Concerning your soil data: please provide standard deviation for all results

**We have included all standard deviations where possible in the data tables included in the paper.**

p. 8, Table 1: please provide results separated for field without intercropping and field with intercropping (additional comment: you provide some of this information later in the text, p.8 27-34… think about re-grouping the information). + probably best to already present Figure 2 here and combine the two.

**We have extended table 1 to include the data averaged over all sampled fields as well as per management type and the accompanying text has been restructured to be more clear.**

p.8, 9: How do you justify your conclusion: "Soil analysis shows that the application of intercropping has a significant effect on the soil nutrient content" I find this conclusion difficult to be drawn from the data you have at hand. Please discuss in more depth on how you come to this conclusion.

**The phrasing of this sentence may have been off. We meant to indicate that, although small, the differences in the soil nutrient content for the two compared management types were statistically significant. We have amended the text to clarify this.**

p. 8, 15: "slightly higher" please quantify
p8, 16-17: Covering: at which period of time? Is it the overall and average covering over the whole growing period? Please be more specific on who you quantified the coverage.
p.8 , 17-18: "manure information" should come earlier, when presenting the sampled plots.
P.8, 27-34: please indicate for all your results which differences were significant! Not only for the results presented in Figure 1

**We did include in the earlier version in the text that all of the differences in soil nutrient content between the two management types were statistically significant. This may not have come forward properly due to the structure of the results section in this version.**

Figure 1: what do the differences you found in Figure 1 tell you? Can the differences be explained by the management practices? What could be the reason for the differences? Please discuss.

**These results are now discussed more in-depth in the discussion section of the paper.**

p.10, 16-20: the conclusion "socio-economic factors as determining" did not really become that clear during the presentation of your results. = Re-structure (see my above comments). (other suggestion combine with info on p11 – 24-29 and p12 1-9)
p.10 21-26: does this paragraph belong in the discussion? I would suggestion: "introduction"
p 11, 1-7: part of your results I suggest
p11, 17-21: Repetition! Check with results part.
P 11, 24-29 combine with statement p10, 16-20
P11, 30-40: some of this information could be moved to the soil analysis. … Again a matter of structure.
P12 , 1-9 (Structure! Should come together with p.10 16-20)

**We have restructured large sections of the text and included your comments in doing so. Thank you for pointing out the cases in which there was repetition that we initially overlooked.**

Conclusion
Here you should come back to your main question: "link women's knowledge on agricultural practices and their motivations in choosing specific practices with the nutrient content of their soils". I would

suggest critically going through your paper, improving the structure and in the end coming back to your opening research question. It is no problem, in case you could not fully answer your research question, however be precise in presenting what you could show, what you couldn't show, which questions remain open…

When looking at the information you present I wonder why the woman intercropping actually did the intercropping? To improve soil fertility or to create additional income?

Or otherwise put: what does the woman know? What is the driving force for their action? Is it missing knowledge or socio-economic pressure?

**Thank you for your advice on this. We have gone through the study results again after the restructuring and used this to update our findings in the conclusions. We have also taken care to indicate where there are still questions remaining to be answered.**

p.12 – 32: where did you show that the "cultural status" of women is important for understanding their action? Maybe I did miss this information. But, I can only remember you citing a study from elsewhere, but no information from your study. So, I would say you couldn't show so. Other possibility: clearly explain what you mean with "cultural status" and how the data you collected supports your conclusion.

**In the section on focus group discussions the cultural restrictions on women in agriculture were mentioned and the example of the prohibition on women owning banana trees was given. This may not have been clear in the previous version of the paper.**

Did you show in which way the women's decision might differ from that of the men? In case you couldn't, then no need to talk about it. Could you show with your data how the access of women to knowledge, contacts, material and capital differs from that of the men? If so, please make these differences clearer in the presentation of your results. In case you couldn't show, then mention it as a short-coming, or do not mention it at all. I can only remember reading somewhere that the women said, they are more open to knowledge and advice (p.6, 32)

**We have removed the section mentioning this from the paper as we felt after review that they may give the wrong impression that we are comparing the practices of men with the practices of women.**

**Thank you again for your careful and extensive review and commenting on our paper.**

**List Major Changes:**

Title and Author list:

1. Change title from: Women's agricultural knowledge and its effects on soil nutrient content in the Nyalenda urban gardens of Kisumu, Kenya, to: Women's agricultural practices and their effects on soil nutrient content in the Nyalenda urban gardens of Kisumu, Kenya – to better reflect article contents
2. Change in author list order to be better reflect contributions

Introduction:
3. Removal of 2nd paragraph

Methods and Materials:
4. Removal of 1st paragraph
5. Move of interview and focus group discussion question lists to supplement

Results - Interviews:
6. Restructuring of content to eliminate repeat information
7. Removal of quotes

Results – Focus group discussions:
8. Switched 1st and 2nd paragraph

Results – Soil Analysis
9. Switched position of 1st and 3rd paragraph
10. Addition of soil data per different management practice described in table 1

Discussion;
11. Removal of repeat information in 3rd paragraph
12. Removal of 5th paragraph

Conclusion
13. Addition of indication for need of future research and clarifying information

Supplement:
14. Addition of supplement containing interview and focus group discussion questions and anonymized participants data

**Article Submission – Track Changes:**

**Women's agricultural  practices and  their effects on soil nutrient content in the Nyalenda urban gardens of Kisumu, Kenya**

Nicolette Tamara R.J.M. Jonkman[1], Esmee D. Kooijman[1], Karsten Kalbitz[2], Nicky R.M. Pouw[32], Boris Jansen[1]

[1] Ecosystem and Landscape Dynamics group, Institute for Biodiversity and Ecosystem Dynamics, University of Amsterdam, Amsterdam, 1090 GE, the Netherlands

[2] Soil Resources and Land Use, Institute of Soil Science and Site Ecology, Technische Universität of Dresden, Tharandt, 01737, Germany

[32] Governance and Inclusive Development Programme Group, Amsterdam Institute for Social Science Research, University of Amsterdam, Amsterdam, 1018 WS, The Netherlands

*Correspondence to*: Nicolette Tamara Jonkman (N.t.Jonkman@uva.nl)

**Acknowledgements**

We would like to sincerely thank all those who aided us in the preparation and execution of this research project. Among whom: the Kisumu VIRED team, including  Professor JB Okeyo-Owour and Dr. Dan Abuto the CABE team in Nairobi, including Dr. Hannington Odame, and NWO-WOTRO for funding of the project (W 08.250.200).

**Abstract.** In Kisumu up to 60% of the inhabitants practice some form of urban agriculture, with just under 50% of the workers being female. On average, women spend more hours a day in the gardens than men. To increase yields, women's knowledge is pivotal to effective agricultural management. This means taking into account that women face greater obstacles in land ownership, investment, and farm inputs due to social and cultural constraints as consequence of their gender. This case study aimed to  better understand the position of women farmers in the urban gardens by determining what agricultural knowledge the women farmers  
[revised manuscript text omitted]

---

## Author Response (AR3)

**Contents**

30

35

40

**Response Reviewer 1:**

**We would like to thank the reviewer for taking the time to read and review our manuscript. Below we have given response to the different comments.**

p.5 L. 40: introduce the abbreviation FGD in the paragraph above. To make understanding easier, use mixed FGD, women FGD, both FGDs. As it is now, it is still a bit confusing, which information is derived from where.

**We received a similar comment from the other reviewer. We have introduced the abbreviation FGD in the methods in this revision (P4, L23) and have changed the relevant paragraphs to show more clearly which information came from which focus group discussion (mixed or women-only) (Paragraphs 1 and 2, Page 7). We hope that this alleviates the confusion created.**

p. 9 L 37. To which information do the references point to? That knowledge passed down from previous generations does not travel far in general, or are the references specific for the Nyelenda group? Please specify: e.g. It was also shown by ..., that

**We see how the term knowledge is too vague and have specified the type of knowledge. We have also specified that this concerns specifically the situation found in the urban gardens in SSA (P11, L10-11).**

Three questions you could include for better understanding:
1. Why did government extension recommend intercropping for increasing soil N, although the soil is rich in nutrients?

**Nutrient recommendations are soil and crop specific, so while a soil may not have enough nitrogen to support the growing of maize, it can support the growth of cabbages or other leafy vegetables. The government extension workers may have made their recommendation based on what was being grown at the time on some of the fields, or on the request of a farmer in the group that would like to grow maize, or based on the comparison of soil analysis results to a standard they are supplied with by the county agricultural office. This is unfortunately something we cannot determine with certainty and also, in our view, does not fall entirely within the scope of our manuscript. As such have not included it in the manuscript.**

2. Can you make conclusions on the practice of intercropping cowpeas and selling the crop, also? Is this practice recommendable, or does it decrease soil nutrients too much? Or is there more research needed, especially also on the effect of not incorporating the manure thoroughly?

**To have more certainty as to the long term effects of the current practice, especially in concurrence with the fertilization practice, more research is needed. We have included this in our conclusions (P13, L11-12). Due to the natural richness of the soils under study, the current practice is not depleting the soil much faster, and more importantly not impacting the yield or crop quality noticeably. We have included a section in the revised manuscript in which this is mentioned (P6, L30-34). Whether the practice can really be recommended would need a close consideration of both the environmental and financial aspects.**

3. What is the main benefit of the soil analysis for the understanding of the interview and FGD results? How would you propose to use the combination of soil analysis and farmer interviews in future (bigger) studies?

The results of the soil analysis help to understand if changes in agricultural practice (i.e. the implementation of intercropping) have an actual influence on the soil nutrient content. The interviews and

5   FGD alone would not have shown that intercropping as it is currently practiced removes more nutrients from the soil than growing only kales.

Also, during the initial field work it was assumed the farmers were practicing intercropping by ploughing the cowpeas into the soil. If only the soil analysis results would have been used without the interviews and the FGD, the conclusion might have been drawn that the technique is ineffective. In this study the two

10   (types of) methods supported one another and led to a more holistic view of the situation.

In future studies similar methods can help scientists develop more suitable agricultural practices, which take into account the socio-economic circumstances of the farmers. As the same time, a larger study can also more easily involve the farmers; this would give the farmers more ownership of both the collected data and the developed agricultural practices. This increase in ownership is likely to increase the likelihood

15   of uptake and implementation of newly developed agricultural practices.

We would again like to thank the reviewer reviewing our manuscript and giving thoughtful comments and questions based thereon.

30

35

**Response Reviewer 2:**

**First of all we would like to thank the reviewer for again taking the time to go through the manuscript so thoroughly and highlighting the weaknesses. We hope that this time we are able to more fully address these issues raised to satisfaction.**

The article improved considerably concerning structure, however, there are still some language issues; especially on the first two pages.

**We are sorry to hear about the language issues, though happy to hear that the structure has improved. To prevent more language issues we had our manuscript checked by a native language speaker before submission this round.**

More important, unfortunately concerning the content the article still remains limited in scope and some more input regarding the information gathered during the interviews would be necessary in order to show that the conclusions drawn are based on the field work. The main aims of the paper to understand "what knowledge women working in urban agriculture have on agricultural practices, and how their practices affect their soil" (p.3, L8-9), & "we aimed to determine how the agricultural knowledge and motivations of women farmers influences their soil' nutrients status" remain so far only sufficiently answered, as no detailed profiles of the two women concerned, their knowledge and their motivations and how they differ are given. Furthermore, the differences in management practices are described very superficial, which makes it almost impossible to say whether the differences in soil properties come from the management practices or rather from the long-term use history or the baseline conditions of the soil.

**We have tried to limit the scope in the previous version as per the reviewer's suggestions. However, this was clearly not extensive enough. We did include more information on the interviews and focus group discussions in the previous version as requested, however these were added mostly in the supplements and therefore not immediately clear. We moved more of the information to the main text now and expanded it even further in an attempt to accommodate the reviewer's suggestions as much as possible. Below we have indicated where one may find the specific information connected to all specific comments of the reviewer. In some cases we were unable to fulfill the request as the data was not available. This is also indicated below, and has now been explicitly mentioned in the text as well.**

1, Concerning the aim of the paper focusing on women knowledge and practices and how it differs from men's knowledge few new insights seem to have been generated through the research, the only reference found with regards to this is that the FGD including men mentioned some other information sources (not clear however whether this statement came for a men) and maybe have better access to mineral fertilizer (?). The study does not present in how far the practices of women really differ from that of men in the case study, here the results from FGDs should be presented in more details.

**While it was our intent to focus on women's knowledge, it was not our intent to directly compare it with men's knowledge; this is outside the scope of this article and this research. As such, in the previous**

**resubmission we attempted to remove any sections that might have given the impression that a comparison between men and women was an aim of our study. It was in the mixed focus group discussions that mineral fertilizers were named more often as a tool and in interviews with the women they indicated that men often keep resources for themselves rather than sharing equally, this led to the conclusion that**

5 **mineral fertilizers are used more often by men (P7, L14-L16). We have rewritten the section on the focus group discussions which will hopefully give a clearer picture of the information gained and from whom the information came. In addition, we have now indicated even more explicitly that a comparison between men and women was not an aim in itself (P3, L16-17).**

10 2, Little effort can be seen to understand the reasons for the two women to decide for the different strategies. So it becomes difficult to understand the decision processes. I cannot see where "the study showed that women are influenced by their socio-economic and cultural statues when making decisions in agricultural management" /p11, L3&4, the detailed reasons are not presented; we do not get any information about why the two women use different approaches. The results of the interviews should present the differences (and or similarities) between the

15 two women for which the soils were sampled.

E.g. How old are the two women, how many people do they have to take care of, what are their specific sources of information, for how long have the practiced the soil management practice, why do they do so. What does the women using intercropping say about the soil fertility on her field? Is she aware of the problem of removing the cowpeas? Or is she simply so far happy with the soil fertility = no need to leave the cowpeas on the field (this

20 should also be discussed in relation to the soil analysis, e.g. would the soil analysis suggest that she would need to fertilize her field or are the values still high enough, making fertilizer use unnecessary) Other examples in this regard:

- p3, L31: here it would be interesting to understand who changed strategies, and why.

**We agree in hindsight that in our attempt at conciseness we included too little information on the two**

25 **women whose fields we sampled in the previous resubmission. We have corrected this by including a table with information that should allow for a clearer overview and easy comparison (P7, table 2) as well as a more in-depth exploration of their specific interviews including the women's view on their practices and its results (P6, L27-35). More specifically, the table includes information on the women's agricultural management, their experience and crop coverage in the fields sampled. Some of the requested information,**

30 **such as family size, we could not provide.**

**We believe the influence of the socio-economic circumstances is demonstrated by the continued intercropping with cowpeas for financial reasons, but may not have linked this clear enough in the text. The cultural limitations for women were shown through the context (P2, L24-37) and the results of the focus group discussions (P7, L11-12), but also here we could have marked these results more clearly in the**

35 **relevant section. We have corrected that in this new version of the manuscript.**

3, The potential and the real impact of the two applied strategies is impossible to discuss, based on only rough qualitative description. The article does not give quantitative information concerning the amount of manure that is applied on the fields, the spacing of the cowpeas, the time that the practices have been different and so on. Or

e.g. on understanding the long-term history of the field. While it is mentioned that one of the fields was left fallow the year before, one was planted with maize and two with kales, the information from one year alone does not really help.

The study could furthermore discuss which kind of input is probably provided through the kind of manure applied and how this differs from the input that cowpeas could bring (even if they are harvested… at least in case the root are left in the soil.

The potential reasons for the differences in CEC should be discussed.

**We agree that it is difficult to draw firm conclusions on some of the data provided. For the most part we were forced to rely on qualitative descriptions as these farmers do not keep track of their inputs and the study was too short a duration to track these ourselves. Wherever data was available, such as the coverage of the vegetables per field, it can now be found in the data table added on page 7 of the manuscript or in the text on page 6, L15-25. The limitations of the study are now more clearly discussed in the manuscript (P11, L29-34) and the conclusions have been amended to better reflect the limitations of the study as well (P13, L7-12). A potential reasons for the higher CEC is included in the discussion (P11, L21-28)**

5, As the findings might be too limited in answering the research question, the focus could move to a closer description on the findings that could be achieved (the FGD in more depth, the interview analysis in more depth (e.g. the daily strategies at least of the two women….).

**We hope that with the revision of the manuscript and the inclusion of the additional data as described in the comments above we have provided this shift of focus.**

Some other specific points:

P.2, Paragraph 3: it is not clear which of the information given (sources cited) refer to Kisumu specifically and which ones talk about women (in which countries?) in general.

**We expanded the text of this paragraph and moved several references to make it more clear which sources cover which location/area.**

P3, L1: Reference SAITO is missing, Is SAITO referring to Kenya, Kisumu, the World?

**We are grateful the reviewer pointed this oversight out. We have added the reference to the list and specified in text what area was covered by the research.**

P.3, L 10-13: should rather be moved to the methods part.

**We have moved all information in this paragraph that is related to the description of the field sites to the methods (P3, L19-24).**

P3, L 20 "of" is missing "one OF the" (there are more language mistakes in the text, this is just one example)

**Thank you for pointing out this error. As mentioned above, we arranged to have a native speaker check the manuscript before re-submission to ensure these errors are removed.**

P.5, L11: Why do you talk about "Some" Women. Given the sentence just before it would make sense to give the concrete number here. IN general, the information about the constraints reported could best be presented in table form (constraint and number of times mentioned)

**We understand and appreciate the need for numbers that are as exact as possible. At the named instance we have now included a specific number (P5, L32-33). We attempted to construct a table as requested, but had to conclude that the data was too limited for this. Therefore, we did include it in the body text and more clearly and explicitly addressed this issue there. In line with this, we also more clearly expressed the limitations of the study in the discussion and conclusion (see our response to the previous comments of the reviewer).**

Like on P5 the information sources and the differences between men and women e.g. Sources of information Number of times mentioned by women Number of times mentioned by men Radio

**As we indicated previously, the focus of our manuscript explicitly does not lie on a gender comparison. We have altered the text to explain this more clearly (see our response to the previous comments of the reviewer).**

P5 L 15-16: Sentence a bit strange and not really useful here.

**The sentence was amended in its present form based on a request of the other reviewer in the previous round. We have attempted to rephrase it.**

P5, L21: Bring examples on how you concluded that the knowledge is limited to visible effects?

**This was directly indicated by 6 of the 8 women during the interviews. During 7 of the 8 interviews it was indicated that they did not know how fertilizers work or what they do in the soil and that they just know that it helps their crops to grow. The one interviewee that did have some knowledge only had a rudimentary understanding of the working of fertilizers. We have altered the indicated section to reflect this more closely (P6, L1-3).**

P5, L34: which resources? Where do they come from?

**We have added an example that was named during the focus group discussion to clarify (P7, L7-8).**

P5, L35f: mention all the cultural restrictions that were mentioned and which ones are still followed and how they impact on soil management practices.

**All examples mentioned in the focus group discussions are already present in the text. We have now explicitly mentioned this fact in the text (P7, L9-11).**

P5, L 38: consider deleting the last half sentence.

**We deleted it as requested.**

P6, L1: discrepancy: a pity you can't say more

**We agree that this is unfortunate, however the limitations of the study, which we now more explicitly delineate, indeed prevent further elaboration.**

P6, L25-34: why do they use different practices? (+ this part does not really belong to the soil analysis, but rather to the interviews)

**We have moved this section and amended it as requested. Some more information on where the practices come from may be found on page 6 (L27-35)**

P9, L19-23: this paragraph should rather be moved to the introduction

**Moved to introduction, incorporated into paragraph 4, page 2 (continues on page 3)**

P9, L24-29: here it should rather be discussed whether differences between men and women could be found or not and to which extent and what can be learned out of that. E.g. are there really important differences in knowledge between men and women? Yes or no, this should be discussed here.

**As mentioned before, it was not within the scope of this study or article to make a comparison between men and women.**

P9, L38-43 Reasoning in this paragraph is wrong, as the example given doesn't provide evidence for the claim that decisions are heavily influenced by socio-economic constraints. Here it would be better to enter the information given p10, L14-22 which gives at least one socio-economic reason. Again it would be interesting to know whether famers are aware of the fact that harvesting reduces the positive effect of the intercropping.

**Upon re-reading we agree that the current example is suboptimal. We replaced it with a better one (P11, L16-21).**

P. 10 L 1-10, might in parts better be moved to chapter 3.3

**We understand that some parts of this paragraph could be interpreted as a result and have attempted to correct this by removing some of these statements. Any remaining statements were kept in the paragraph to preserve the flow of the argument (P11, Paragraph 2).**

P10: L8 and 9 Numbering of farmers (I2,I7) has not been introduced before. No need to introduce it at the end. See comments above, here it would be interesting to get some information about why one of the farmers is ploughing the manure into the soil and the other doesn't. It would further be interesting to know, what "ploughing" means (3cm, 10cm, 20cm?). .

**Numbering has been introduced earlier and with better context (Table 1 on page 4, and paragraph 2, page 6). Ploughing is done by handheld tools that reach an approximate depth of 20 cm; we have included this in the manuscript (P6, L19).**

P10, L24-26: Am not convinced that the study could show this. Yes, there are differences, but do they come from this one year of doing either only manure or manure and intercropping? Maybe in the past the management practices have been much more different? Maybe one of the soil samples contained more sand than the other?

5  (Did you measure the clay, loam, soil content?)

**We have changed the text to reflect the uncertainty better (P12, L13-18). We do not have multiple year observations and now more clearly indicate that for full quantification it would be necessary to focus on land-use history over longer time-scales. That said, the lack of a positive effect coinciding with the improper use of intercropping (i.e. crops were not ploughed into the soil) do make our observation of a**

10  **lack of a clear positive effect plausible. In addition, from the interview results we know that the farmer started intercropping in 2013 and has been applying this practice ever since. The fields were sampled in 2016, which means the practice had been going on for 3 years. At the same time the other farmer decided not to adopt te practice, which means the field's treatments started to deviate 3 years ago. The results are the net effect of those three years of different management. This is now more clearly indicated in the text**

15  **(P6, L23-25).**

L26: Is it really the growing of cowpeas that causes a more rapid extraction from the soil? (What is the plantation density?). It could also be that farmer that is only applying manure is simplying applying more manure. I would be careful with such statements.

20  **Due to the inclusion of the cowpeas the fields that are intercropped have a greater plantation density. This information can now also be found in the data table on page 7. However, it is indeed possible that manure amounts and methods may have had a significant effect; we have included this in our discussion (P11, L16-27).**

25  L29-31: Given the scope of the study, it would really make sense here to talk to the farmer again and ask her, why and when she decided to intercrop and what she thinks about the strategy.

**The answer is already available in the interview data, but clearly was not made explicit well enough by us in the previous versions. The farmer adopted intercropping with the others that did in 2013 after a visit by local extension workers that advised them to do so. She has kept up the strategy as she has noticed that it**

30  **gives her financial benefits, in the manner that she currently practices it (P6, L28-33).**

P11, L4-6: "The case study showed that women are influenced by their socio-economic and cultural status when making decisions in agricultural management and these decisions may differ from those of men in the same or similar circumstances due to a lack of access to knowledge, contacts, or material and capital." I think it would

35  help a lot, if the authors think about this conclusion and bring together all specific statements (in the interviews and the FGD) that support their conclusion. As said before: what are the specificities of the socio-economic and cultural status of women in Nyalenda that were found during the study and that influence management decisions and how do they differ from that of men and why; which kind of different contacts, access to material and capital

of men could be found on the ground compared with that of women? Please provide us with the kinds of questions you asked during interviews and FGDs

**We did include this information in the previous resubmission. However, we had placed it in the supplement for the sake of conciseness. It is clear that we failed to clearly indicate the availability of this information in the main text. We have corrected this and now more clearly and explicitly refer to the additional information in the supplementary material where appropriate in the format that is laid out in the authors guide of SOIL.**

**We would like to thank the reviewer again for their thorough and clear review of our manuscript. Their contribution has helped us increase the quality of the article.**

**List of major changes:**

- Abstract: revised
- Introduction: moved section from last paragraph to methods chapter
- Methods: inserted table 1: list of interview participants
- Results: inserted table 2: Details on sampled fields and related management practices
- Results: moved paragraph on agricultural practices sampled fields from soil section to interview section
- Results: inserted paragraph elaborating on management practices farmers whose fields were sampled
- Discussion: inserted elaboration on effects of soil management on soil characteristics
- Conclusion: revised first paragraph

30

35

**Track changes version of the manuscript:**

**Women's agricultural practices and their effects on soil nutrient content in the Nyalenda urban gardens of Kisumu, Kenya**

Nicolette Tamara R.J.M. Jonkman[1], Esmee D. Kooijman[1], Karsten Kalbitz[2], Nicky R.M. Pouw[3], Boris Jansen[1]

[1] Ecosystem and Landscape Dynamics group, Institute for Biodiversity and Ecosystem Dynamics, University of Amsterdam, Amsterdam, 1090 GE, the Netherlands

[2] Soil Resources and Land Use, Institute of Soil Science and Site Ecology, Technische Universität of Dresden, Tharandt, 01737, Germany

[3] Governance and Inclusive Development Programme Group, Amsterdam Institute for Social Science Research, University of Amsterdam, Amsterdam, 1018 WS, The Netherlands

*Correspondence to*: Nicolette Tamara Jonkman (N.t.Jonkman@uva.nl)

**Acknowledgements**

We would like to sincerely thank all those who aided us in the preparation and execution of this research project. Among whom: the Kisumu VIRED team, including Professor JB Okeyo-Owour and Dr. Dan Abuto, the CABE team in Nairobi, including Dr. Hannington Odame, and NWO-WOTRO for funding of the project (W 08.250.200).

**Abstract.** In Kisumu up to 60% of the inhabitants practices some form of urban agriculture, with just under 50% of the workers being female. On average, women spend more hours a day in the gardens than men.  Therefore women's knowledge is pivotal for effective agricultural management. To enhance and better use women's knowledge, gender related social cultural obstacles linked to land ownership, investment, and farm inputs have to be taken into account. We aimed to determine how the agricultural knowledge and motivations of women farmers working in the Nyalenda urban gardens in Kisumu (Kenya) influence the soil nutrient status as reflected by the total soil C and N, available soil N and P and exchangeable soil Na, K, Mg and Ca. ~~This case study aimed to better understand the position of women farmers in the urban gardens by determining what agricultural knowledge the women farmers 
[revised manuscript text omitted]